


a

# Discharge and sediment fluxes along the Amazon river: RDSM model concepts and validation

Safaa Naffaa[1], Frances F.E. Dunne[1], Jannis Hoch[1], Geert Sterk[1,†], Steven S.M. de Jong[1], and L.P.H.(Rens) van Beek[1]

[1,†]deceased
[1]Physical Geography department - faculty of Geoscience - Utrecht university, Netherlands

**Correspondence:** Safaa Naffaa (s.naffaa@uu.nl)

**Abstract.** The Amazon covering approximately 6.8 million $km^2$, is the largest river by discharge in the world, and transports copious suspended sediment. In the Amazon, human activities such as dam construction and land use change are likely to affect sediment transport strongly as the basin moves further away from pristine conditions.

In this study, we applied the River Discharge and Sediment model RDSM to simulate annual and monthly discharge and sediment transport in the Amazon between 1980 and 2009. To this end, the model couples sediment production with runoff generation and river transport. The model works at a spatial resolution of 5 arc minutes. It accounts for the impacts of land use change on runoff generation and sediment production, and for the entrapment of sediment by lakes and reservoirs.The sediment load in the stream of every cell at every time-step, therefore, reflects sediment production, uptake and deposition as it is transported and accumulated along the drainage network.

We validated the model using the Hybam-project dataset for seven discharge stations distributed over the Amazon. Additionally, estimations of sediment transport from previous studies were used as benchmark. The model is able to effectively capture the monthly and annual variations of discharge with Kling-Gupta Efficiency ranges from 0.57 to 0.92 and sediment transport within the basin and to the ocean with Kling-Gupta Efficiency ranges from $-1.7$ to 0.49.

Based on the model results, the annual average sediment transport (1980-2009) at station Obidos Porto, the station furthest downstream, is $6.46 \times 10^8$ tonne/yr and the annual average discharge is $17.5 \times 10^4$ m$^3$/s.

The model estimates the annual average sediment transport to the ocean at $5.96 \times 10^8$tonne/yr, which is in the same order of magnitude as field measurements and is in line with the results of other studies.

The RDSM model facilitates future estimation of sedimentation impact in reservoirs incorporating water resource management and will so contribute to a better understanding of the complexity of the Amazon basin.

## 1 Introduction

Rivers are the main suppliers of sediment to the oceans. The Amazon is one of the largest rivers globally (Martinelli et al. (1989a)), covering approximately 6.8 million $km^2$ and containing about 20% of the world's fresh water (Meade et al. (1985); Filizola Jr and Guyot (2009); Manning (2011)). The Amazon river transports large quantities of suspended sediments (Anthony





et al. (2010); Manning (2011); Latrubesse et al. (2017)). The large volumes of water and sediment affect the global water cycle and nutrient balance (Martinelli et al. (1989b)), and the exchange of sediment between the land and the ocean influences the global carbon cycle (Martinelli et al. (1989b); Langerwisch et al. (2015)). The Amazon forest also plays a crucial role in the global climate system by absorbing solar energy and recycling half of the regional rainfall (Marengo et al. (2011)) but it is threatened by deforestation, which could heighten the export of suspended sediment and nutrients to the ocean, and the

hydrological response as a result of climate change. The sediment transport along the river is vital for, among others, agriculture activities downstream and the survival of important fluvial and coastal ecosystems. Furthermore, sediment transport to the ocean determines rates of coastal erosion. Therefore, studying sediment production in terms of the supply to the Amazon and its subsequent transport along the Amazon Basin is vital to understand the dynamics of sediment production in local agro–ecosystems, sediment delivery and the stability of the coastal zone, and feedbacks within the global climate system influencing

sediment transport (Liang et al. (2020)), as well as the interactions between these components.

Efforts to quantify the suspended sediment flux at the mouth of the Amazon range between $5\times10^8$ and $13\times10^8$ tonne/yr. Gibbs (1967b) was the first to estimate the suspended sediment flux at $5\times10^8$ tonne/yr based on 74 surface samples on two cruises during low and high flow periods. A later study by Oltman (1968) used only three surface samples and yielded a higher value of $6\times10^8$ tonne/yr. Meade et al. (1985) produced a new estimate of $11\times10^8$ tonne/yr to $13\times10^8$ tonne/yr based on sam-

ples collected from researchers of the the Carbon in Amazon Research Experiment CAMREX project. They collected frequent single cross section samples every four months between 1982 to 1984 (Filizola Jr and Guyot (2009)). The Alpha-Helix project estimated the mean suspended sediment flux at station Obidos Porto, the most downstream station at $9\times10^8$ (Mead and Curtis, 1979). Their estimation was based on 325 samples collected from several points in repeated cruises during the high flow seasons for one year (1976-1977) between Peru and Brazil and from the lower reaches of most of its major tributaries. In the

1990s, the Amazon shelf sediment study AmasSeds used transmissometer equipment from 190 stations to estimate sediment transport at the mouth of the Amazon river at $5.50 - 10.30\times10^8$ tonne/yr (Filizola Jr and Guyot (2009)).

To complement the limited field measurements, computer models have been used to estimate sediment transport in Amazon river, such as the continental sediment model MGB-SEDAS developed by Fagundes et al. (2023) who arrived at $4.06\times10^8$ tonne/yr near the mouth of Amazon. Hatono and Yoshimura (2020) used a global sediment dynamics model to sim-

ulate the mean annual suspended sediment transport at 60 stations in South America, Africa and Europe. Their simulations for the stations of Obidos Porto,Manacapuru, Portovelho and Serrinha stations in the Amazon basin (see also Figure2), were $4.01\times10^8, 0.9\times10^8, 2.3\times10^8$ and $0.3\times10^8$ tonne/yr. Hoch (2014) estimated the annual sediment transport near the Amazon mouth at $37\times10^8$ tonne/yr using PCRGLOB-SET model. Finally, Pelletier (2012) estimated long–term sediment transport for the Amazon basin, giving a sediment yield of $140$ tonne/km$^2$/yr, which approximately corresponds to $9.5\times10^8$ tonne/yr

of sediment transport.

Although the estimated and modelled values partly overlap, they also vary widely, highlighting problems of data scarcity on the one hand, and of scaling, process identification and parameterization on the other. Unfortunately, sediment production and transport modelling for the Amazon basin is a challenging task due to the complex nature of the soil erosion and sediment transport process. To better understand these processes and assess the impact of human activities on sediment transport, spatial-





temporal process-based models are required capturing all relevant processes. We tried to cover all relevant scales and processes, bearing in mind the possible impact of global change (e.g., land cover and climate), and simulate the transport of suspended sediment from the source to the basin outlet. To this end, we present the River Discharge and Sediment Model RDSM.

We assessed the spatial-temporal distribution of discharge and sediment transport within the Amazon basin and to the ocean over the period 1980-2009 at 5 arc minutes. We used this model to explore the basin-wide historical development in light of

past changes considering the influence of historical land cover and climate. We analyzed the amount of erosion produced due to the hillslope and delivered to the river per catchment and per catchment area. We validated it against the Hybam observations of discharge and against the Hybam observations and previous studies' simulations of suspended sediment transport. We used Kling-Gupta efficiency KGE and the relative value of Root Mean Square Error RMSE to validate the performance of the model.

## 2    Study area

The Amazon basin (Figure1) is shared between seven countries in Latin America: Brazil, Bolivia, Peru, Colombia, Ecuador, Venezuela and Guyana. It is home to the largest rainforest in the world, which covers two-thirds of the Amazon River basin. This forest is rich in biodiversity (Anthony et al. (2021); Foley et al. (2007)), and plays an important role in regulating the climate by its role in water, energy and carbon cycles (Marengo et al. (2018)).

The Amazon River basin has a hot and humid tropical rainforest climate (Kottek et al. (2006)). The average temperature

ranges from 25.9 C$^\circ$ (January to April) to 27.6 C$^\circ$ (August to November) (Fraser and Keddy (2005)). Its tropical location results in a high annual rainfall, which follows a seasonal pattern that varies spatially between 3000 mm in the west to 1700 mm in southeast. The wet season differs between the northern, western and southern region of the Amazon basin, running from April to August in the northern Amazon and from January to May in the western region while it runs from October to April in the southern Amazon (Ronchail et al. (2002)). Due to high rainfall volumes and its vast basin area, the Amazon river has a

high discharge of $6.5 - 6.6 \times 10^3$ km/yr (Moquet et al. (2016)). Rio Madeira and Rio Negro are the largest tributaries with similar average discharge of approximately $10^3$ km$^2$/yr (Martinelli et al. (1989b)). An estimated 90% of the total suspended sediment carried to the Amazon river originates from the Andes mountains due to intense erosion caused by the large volumes of high-intensity rainfall and their pronounced topography (Wittmann et al. (2011); Meade et al. (1985)). Hence, the Rio Madeira, which drains part of the Andes, delivers approximately 50% of the total suspended sediment, while the Rio Negro

carries negligible amounts of suspended sediment (Ayes Rivera et al. (2021); Park and Latrubesse (2014); Filizola Jr and Guyot (2009)).

## 3    Methods & Data

RDSM (Figure 2) is a spatial–temporal process–based model, in which the model domain is discretized by a regular geographical grid in WGS84 with a spatial resolution of 5 arc minutes (around 10 km to 10 km at the equator), coded with PCRaster

python. At its core is the large-scale hydrological model PCRGLOB-WB 2 (Sutanudjaja et al., 2018). Using the conceptual-



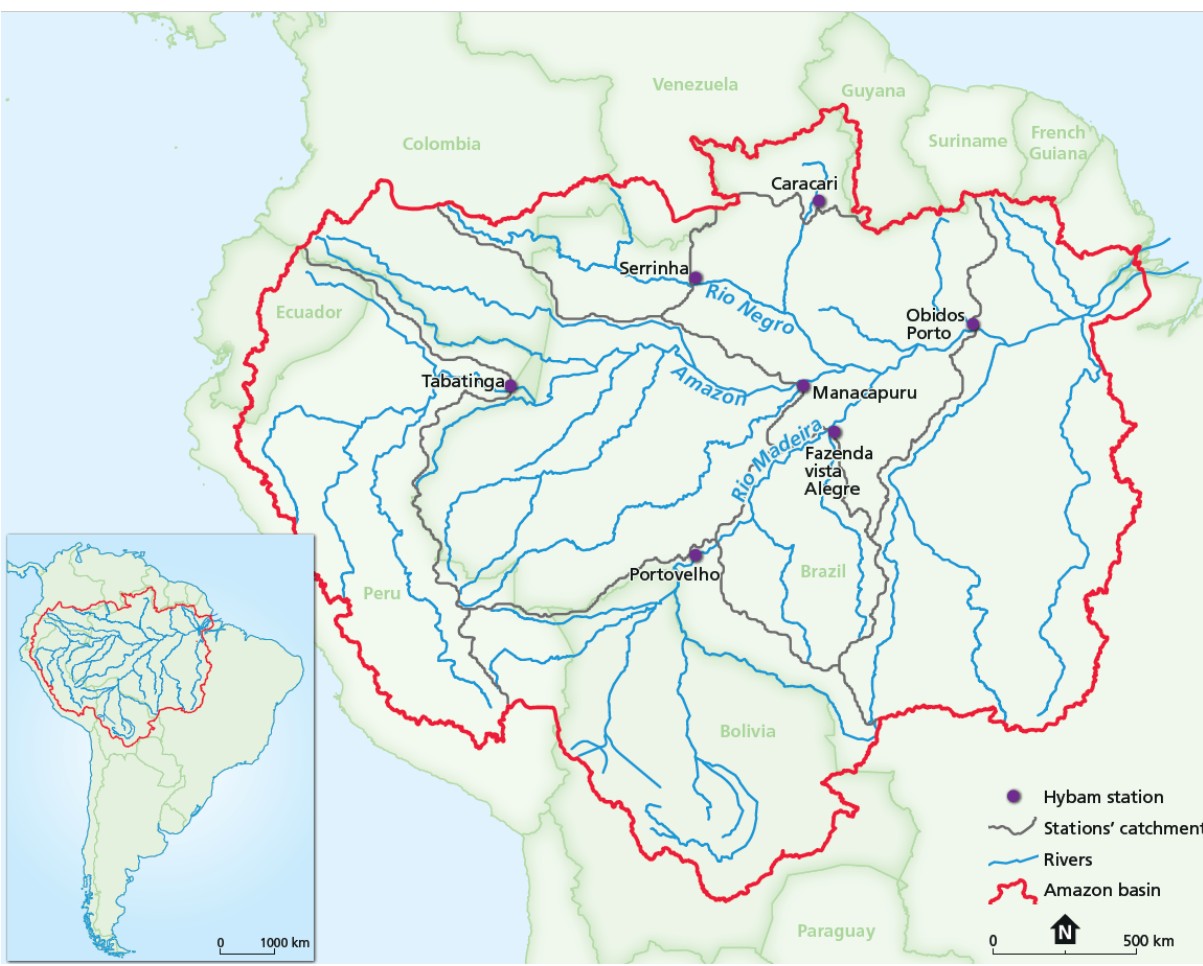

**Figure 1.** Map of the Amazon basin with major tributaries and the seven monitoring stations used in this study from the Hybam project and their catchments. The land cover types per sub basin is shown in Figure F1

ization of the sub-grid variability ofPCRGLOB-WB, each cell can have multiple land cover types that are represented as a fraction. Per cell, while topographical properties such as slope length and slope angle, and the soil properties are derived from data sources at a higher resolution. The temporal resolution of the model is daily and meteorological conditions are imposed from available global forcing datasets which fields can be downscaled using the topography and lapse rates to match the model

resolution. To simulate sediment transport, two model components have been paired to PCRGLOB-WB: the River Sediment Production Model RSPM that is based on the global sediment model PCRGLOB-SET is used to compute the sediment production at the hillslope scale and the sediment delivery to the streams; and the River Sediment Transportation Model RSTM that is linked to the routing model of the global hydrological model PCRGLOB-WB. The human influence on the hydrology, soil erosion and sediment transport can be included by the model: climate change and land cover changes can be imposed and

the effect of reservoirs ( Figure 3) evaluated in the routing scheme. Combined, the model captures the processes of sediment





production, sediment delivery, and sediment transport and the human interference as they propagate from the hillslope to the river in the Amazon Basin (Figure 2).

## 3.1 PCRGLOB-WB

PCRGLOB-WB (van beek and Bierkens (2009); Sutanudjaja et al. (2018)) is a large-scale hydrological model combining

hydrological processes with aspects of water resources management. The water balance computations at the core of the model describe how on a daily basis precipitation passes through the canopy and how it is split into surface runoff and infiltration; the infiltrated water is converted into soil moisture and can be lost to evapotranspiration or percolate towards the groundwater; once passed to the groundwater reservoir, the water will be converted to baseflow (Sutanudjaja et al. (2018)). While states and fluxes are aggregated to cell level, they are evaluated per land cover type in order to represent the vertical exchange

of water between the atmosphere, vegetation and soil in detail. This allows PCRGLOB-WB to efficiently include sub-grid variability in its calculations for the land surface (see above), which makes it extremely suited to couple it with the RSPM model to compute the sediment production at the hillslope scale, i.e., the amount of on-site erosion, and the sediment delivery to the stream, i.e., the amount of eroded sediment entering the stream. Total runoff is delivered to the surface water, which is subdivided into river segments, lakes and reservoirs that constitute the drainage network, where it is routed as river discharge

using the kinematic wave approximation of the Saint-Venant equation assuming one-dimensional open channel flow (van beek and Bierkens, 2009; Sutanudjaja et al., 2018). When the bank full capacity of the channel is exceeded, river water spills onto the floodplain and reduces the flow velocity. Similarly, the outflow of water bodies of lakes and reservoirs is dampened, and the lower flow velocities affect sediment deposition and uptake in the channel in the model. The water resource model of PCRGLOB-WB (Sutanudjaja et al. (2018)) interacts on a daily time step with the physical hydrology; human water demands

include those of the domestic, industrial and agricultural sectors, the latter comprising both livestock, and irrigation water demand. All demands are imposed as an external forcing with the exception of irrigation water demand that is computed inline to reflect the actual meteorology and soil water availability over irrigated areas. Water demands are allocated dynamically to the available water resources, the main ones being surface water in rivers and water bodies and the groundwater. Potential water withdrawals match the demand with the long-term availability, which is evaluated over pre-defined allocation zones.

These withdrawals are then imposed on the actual available resources, and any unmet withdrawal from the surface water is taken from the groundwater if this water is physical available and its withdrawal is not limited by the pumping capacity. Water withdrawals are partly consumed, the remainder is returned to the hydrological system. For irrigation, the water applied re-enters the hydrological system if it is not lost to evapotranspiration. For all other sectors, it is passed back to the surface water, irrespective of its origin. Dams and reservoirs are human-made structures that have a further important impact by modifying

the river discharge. In PCRGLOB-WB reservoirs are included as water bodies, like lakes, for which the outflow is a function of the storage. For reservoirs, this function is defined by a set of reservoir operations, which by default mimic a dam operated for hydropower generation. PCRGLOB-WB is parameterized with global datasets and it can therefore be readily applied in data-scarce environments such as the Amazon Basin (see for details Sutanudjaja et al. (2018) ) and used as a basis for this study. PCRGLOB-WB has also already been used extensively for the Amazon in similar applications, including assessments





of the hydrological effects of land cover change, land use management and climate variability (Staal et al., 2020; Duden et al., 2021; Schaik et al., 2018), and the hydrodynamics of its river regime (Hoch et al., 2017). Hoch et al. (2017) found that the simulated hydrology is highly sensitive to the precipitation input. For the Amazon, existing global datasets often have a dry bias, leading to underestimation of the discharge. In order to remove this bias, precipitation data were taken from the Hybam database, which provides daily raster precipitation maps with a $1 \times 1°$ spatial resolution. The data used in this study is presented in table F1.

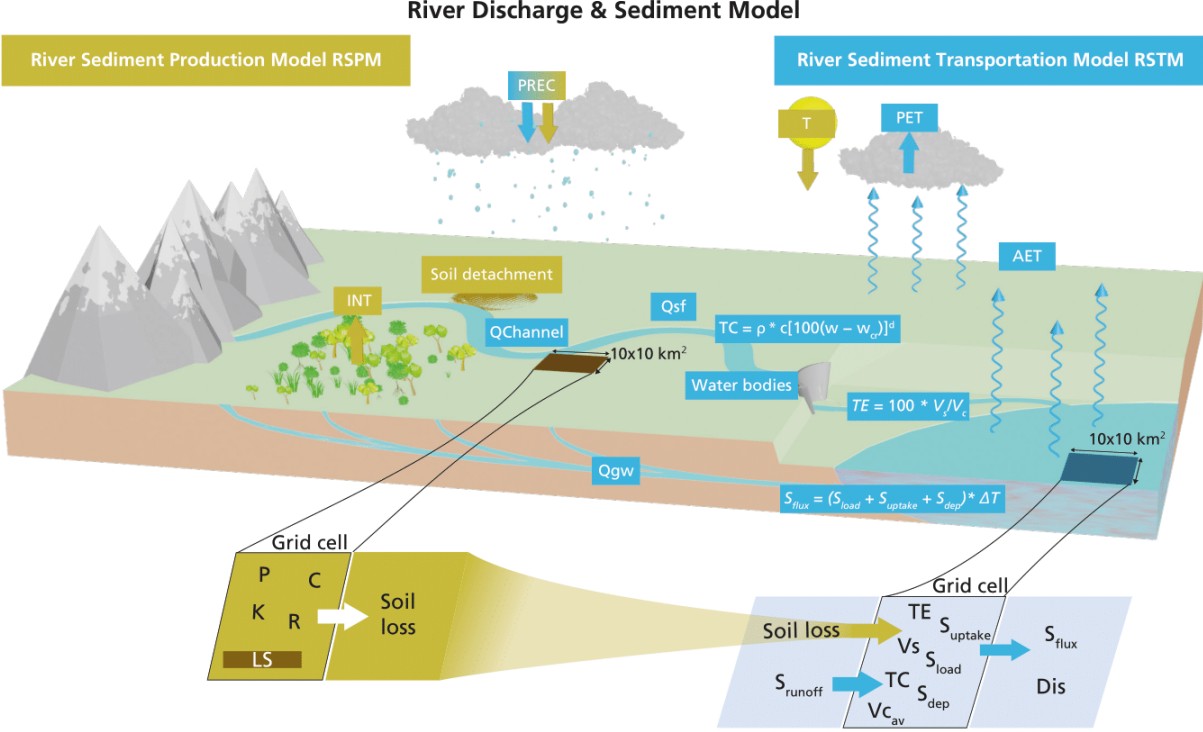

**Figure 2.** RDSM flowchart showing the process of soil erosion and sediment transport in both the components of the model. The River Sediment Production Model RSPM estimates the soil erosion and sediment delivered to the river. The elements used in this model use orange boxes and arrows. The River Sediment Transportation Model RSTM estimates the sediment transport along the Amazon river. The elements used in this model use blue boxes and arrows. The soil loss per area is calculated, using the Revised Universal Soil Loss Equation RUSLE (P: supporting practice factor, C: cover management factor, K: soil erodibility, R: rainfall-runoff erosivity, LS: slope length and steepness factors,INT: Interception and evaporation, PREC: precipitation, T: temperature) (Eq.1). The fraction of transported sediment to the river is calculated using (Eq.4). The total runoff includes surface runoff, direct runoff and groundwater flow is calculated from the daily rainfall. The $S_{flux}$: sediment transported through the river and TE: trapping efficiency of the reservoirs is calculated using (Eq.6 and Eq.16) respectively (PET: potential evapotranspiration, AET: actual evapotranspiration, $Q_{sf}$: surface runoff, $Q_{gw}$: ground water flow, $Q_{channel}$: local discharge along the channel, TC: Transport capacity, $V_s$: settling velocity, $V_{cav}$: critical settling velocity, $S_{runoff}$: surface runoff, $S_{uptake}$: sediment uptake, $S_{load}$: sediment load, $S_{dep}$: sediment deposition, Dis: discharge, $\rho$: particle density, w: unit stream power, $w_{cr}$: unit stream power critical and threshold value, c and d: empirically derived coefficients).





## 3.2 River Sediment Production Model RSPM

RSPM provides the input of sediment to the river due to erosion. It is based on the global sediment model PCRGLOB-SET (Hoch, 2014), which was selected on the grounds of its appropriate process description and its close affinity with PCRGLOB-WB from which it derived part of its parameterzation and hydrological input. RSPM simulates the sediment production, i.e., the local soil loss that is delivered from the hillslope to the river system. To this end, RSPM uses the soil loss and sediment delivery equations from PCRGLOB-SET. PCRGLOB-SET uses the Revised Universal Soil Loss Equation (RUSLE) at a monthly time step ((Renard et al., 1991)):

$$A = L * S * R * K * C * Y \tag{1}$$

A = soil loss [kg/month], L = the slope length [m], S = slope steepness [-], K = soil erodibility [tonne ha h ha$^{-1}$ MJ$^{-1}$ mm$^{-1}$], R = the rainfall-runoff erosivity [MJ mm ha$^{-1}$ h$^{-1}$ m$^{-1}$], C = the cover management factor [-]; P = the supporting practice factor [-].

The slope length factor L (Eq.A1) is constant and reflects the impact of slope length on erosion. The slope steepness factor S reflects the impact of slope angle on erosion and has more impact on erosion than slope length (Renard, 1997). The R factor (Eq.A5–A4) is the monthly erosive energy of rainfall and the resulting runoff (Hoch, 2014). The K factor (Eq.B1–B4) expresses the susceptibility of soil to erosion. The $K$ value differs based on soil type such as clay soil (low $K$ value 0.05-0.15), sandy soil (low $K$ value 0.05-0.2) and silt loam soil (moderate $K$ value 0.25-0.4). Soils with high silt contents have high $K$ values ($> 0.4$) and have less resistance to erosion (Institute of Water Research (2002)). The C factor represents erosion reduction due to vegetation cover and management. Lastly, the P factor is defined by Renard and Freimund (1994) as "the ratio of soil–loss with a specific support practice as compared with soil under unit–plot conditions". Both the C and P factors are dimensionless with values from 0.0-1, where 0 represents no erosion (maximum protection) while 1 represents maximum erosion (no vegetation cover or support practices) (Chen et al., 2019).

The calculations of all factors of the RUSLE equation follow the methods proposed by Hoch (2014), except the C, R and P factors.The P factor was set to 1 due to the unavailability of soil conservation practice and land management data. The C, R were modified to better incorporate the effect of land cover and to resolve the impact of projected land use changes in future work. These modified factors are discussed in detail below. All other factors of the RUSLE are described in Appendix A.

The R factor was modified to obtain the monthly value from the annual value. We followed the procedure of (Hoch, 2014) and scaled the erosivity on the basis of the fraction of the monthly precipitation $P_m$ over the yearly precipitation $P_{yr}$ (mm) multiplied with the annual erosivity R ($R_{yr}$) in each grid cell. To account for the effect of interception on the erosivity, the monthly precipitation was replaced by the effective rainfall $P'_m$ using the interception fraction $Int$ $(-)$ of the Morgan, Morgan and Finney model (Eq.2) (Webster, 2005).

$$P_{m'} = (1 - Int) \times P_m \tag{2}$$





The interception factor is constant over time and depends on the land cover type (TableB1). The $Int$ per cell was calculated as a weighted average based on the fraction of land cover types within the cell.

The monthly C factor is estimated using the monthly ground cover fraction per land cover type ($GC_i$) and using the equation of Yang (2014):

$$C_i = \exp[-0.799 - 7.74 \times GC_i + 0.0449 \times GC_i^2] \tag{3}$$

Then the monthly C factor per cell is calculated as a weighted average of the $C_i$ values using the fraction of land cover type (i) in the cell. The ground cover fraction is taken from the parameterization of PCRGLOB-WB.

The sediment delivery equation used in RSPM quantifies the amount of the soil loss that is delivered from the hillslope to the stream in each cell and reflects the transport capacity which depends on the flow velocity of the surface runoff, the slope

angle and surface roughness. The sediment delivery to the stream is obtained by multiplying the monthly soil loss $A$ with the monthly delivery ratio $DR_m$, the latter being defined in PCRGLOB-SET as

$$DR_m = \tau \left( \frac{\mathrm{H}_{c,m}\sqrt{s}}{n_m l} \right)^{\zeta} \qquad 0 \leq DR_m \leq 1 \tag{4}$$

where $\tau$ and $\zeta$ are empirical parameters equal to 9.53 and 0.79, respectively (Hoch, 2014). The maps of s and $l$ used to compute the L factor in the soil loss module are also used in this calculation (Appendix A). Temporal variations are represented

by varying potential to generate surface runoff represented in hydrologic coefficient $H_c$ (Eq.5) and by agricultural practices represented in Manning's roughness coefficient n [s/m$^{\frac{1}{3}}$] (Appendix A). The hydrologic coefficient $H_c$ was introduced because it was not possible to physically model overland flow when using a monthly time step and a grid cell size of 1 [km$^2$] (van dijk, 2001). Due to this, $H_c$ is a relative value, proportional to surface runoff $Q_s$ [mm] and rainfall $P_{m'}$ [mm]. PCRGLOB-WB computed surface runoff based on daily rainfall data of Amazon basin (see Model input data). $DR_m$ of each grid cell is

accumulated for each month and the entire grid.

$$H_{c,m} = \frac{Q_{s,m}}{P_{m'}} \qquad 0 \leq H_{c,m} \leq 1 \tag{5}$$

### 3.3 River Sediment Transport Model RSTM

The River Sediment Transport Model RSTM is an extension of the routing module of PCRGLOB-WB that reports the discharge and the associated sediment transport on a daily time step. Sediment transport is reported in terms of the actual sediment load

moving along the drainage network and the sediment concentration of the surface water storage of the river segment in the cell. For lakes and reservoirs, these quantities are reported for the water body as a whole. The distinction between lakes and reservoirs based on their capacities was not made, resulting in the inclusion of all water bodies in the analysis. Nevertheless, these water bodies remained consistently present over time, as shown in Figure 4. The sediment delivery to the stream of RSPM is downscaled to the daily input and the sediment mass added to the total sediment present in the river segment. In RSTM. the



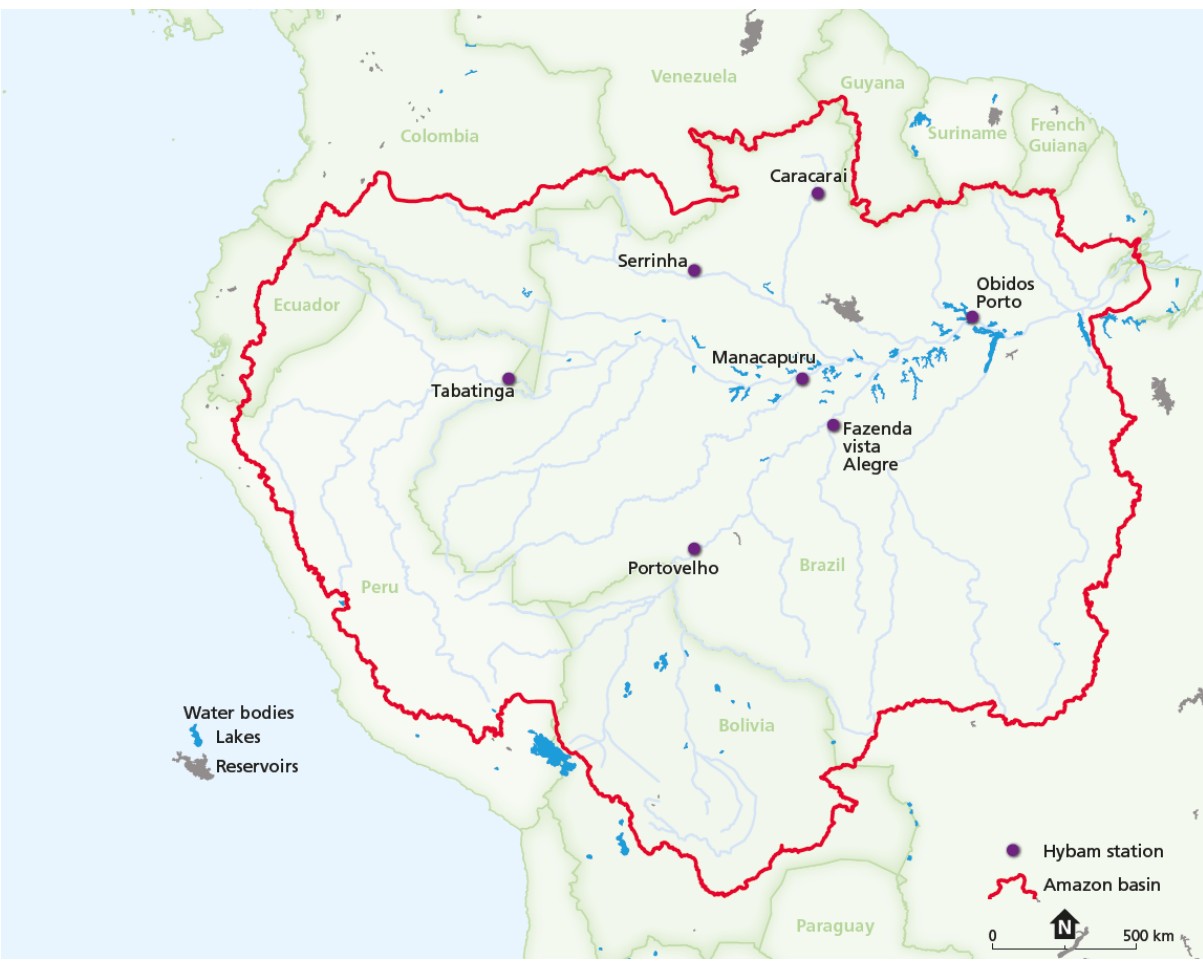

**Figure 3.** Map of lakes and reservoirs (Messager et al., 2016)

amount of sediment in the cell in terms of suspended sediment concentration is compared to the transport capacity TC, which is based on discharge and flow velocity. If TC is smaller than the concentration, deposition occurs as a function of the settling velocity of the suspended sediment. If the TC is larger than the present concentration, additional sediment can be eroded from the river bed, if available, and is added to the suspended load. In water bodies such as lakes and reservoirs, the turnover of the volume by the outflow is small and the residence time consequently large. This results in slow flow velocities over the water body and suspended sediment will settle and is trapped. RSTM incorporates this aspect by means of the sediment trapping efficiency of water bodies, using the approach described by Zaremba (2018). In this study, all water bodies are considered exorheic, so to be part of the contiguous drainage system, and have an outlet.

The sediment transport $S_{tot}$ [kg] is calculated for each cell and every time step using the following equation:





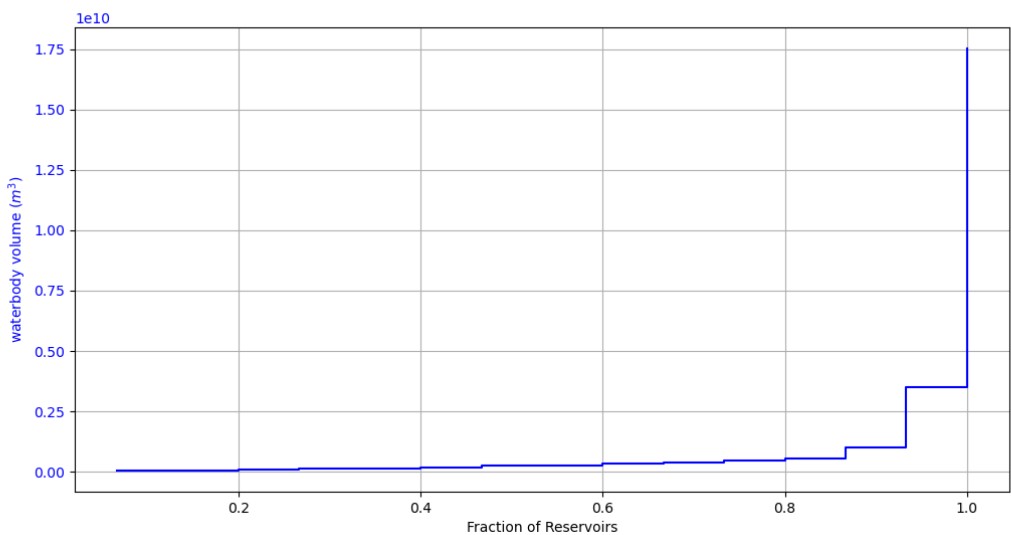

**Figure 4.** This figure shows both the cumulative distribution of reservoir capacities and the corresponding fraction of the total number of reservoirs. The curve rises steeply at first, indicating that a large fraction of reservoirs have small capacities, and it levels off as the capacity increases, showing that fewer reservoirs have large capacities. According to the HydroLakes dataset (Messager et al., 2016), the Amazon contains 14397 water bodies (lakes and reservoirs) with a total capacity of $19.3 \times 10^{10}$ m$^3$. Of these water bodies, only 15 are reservoirs with a total capacity of $2.5 \times 10^{10}$ m$^3$.

$$S_{tot} = (S_{load} + S_{uptake} + S_{deposition}) \times \triangle t \qquad (6)$$

The sediment load $S_{load}$ [kg] is calculated for each cell and every time step $\Delta t$ using the relation between sediment concentration $C_{sed}$ [kg/m$^3$] and water volume $V$ [m$^3$] of the cell (water volume in the channel) (Eq.7). All water and sediment contained by water bodies is assigned to the outflow point of the water body and the remaining cells set to zero.

$$S_{load} = C_{sed} \times V \qquad (7)$$

     The sediment uptake [kg/s] in every cell occurs if the sediment concentration is lower than the transport capacity, and

sediment deposition will occur if the sediment concentration is higher than the transport capacity. The transport capacity TC [kg/m$^3$] is obtained using the method proposed by Govers et al. (1990) (8). It requires as input the particle diameter D [m], particle density $\rho$ [kg/m$^3$], unit stream power w [m/s] and its critical and threshold value $w_{cr}$ [m/s], and two empirically derived coefficients c,d.





$$TC = \rho \times c[100 \times (w - w_{cr})]^d \tag{8}$$

The critical unit stream power indicates the unit stream power below which no transport can occur. The value of 0.007 [m/s] is assumed to be the threshold unit stream power based on Smith et al. (1995).

     The rate of sediment uptake $S_{uptake}$ [kg/s] differs between channel, reservoirs and lakes. For this study, the sediment uptake [kg/m$^3$] in water bodies is assumed to be zero, while in the channel it is determined by the sediment uptake factor $SUF$, the discharge and the difference between $TC$ and $C_{sed}$.

$$S_{uptake} = SUF \times (TC - C_{sed}) \times Q \qquad\qquad \text{if } TC - C_{sed} > 0 \tag{9}$$

$$S_{uptake} = 0 \qquad\qquad \text{if } TC - C_{sed} < 0 \tag{10}$$

     $SUF$ is assumed to be equal to 1 as reworked sediment deposited in the channel should be amenable to erosion and uptake. The sediment deposition in the channel is dependent on the difference between TC and $C_{sed}$, settling velocity $V_s$ and channel width and length (Eq.11), while in water bodies it is dependent on the trapping efficiency and the incoming sediment load

(Eq.13).

$$S_{deposition} = (TC - C_{sed}) \times V \times W_{channel} \times L_{channel} \qquad\qquad \text{if } TC - C_{sed} < 0 \tag{11}$$

$$S_{deposition} = 0 \qquad\qquad \text{if } TC - C_{sed} > 0 \tag{12}$$

$$S_{deposition} = -TE \times S_{load} \tag{13}$$

     To estimate the trapping efficiency TE [-] of the reservoirs, Camp (1945) method is used as it outperformed other methods

according to (Zaremba, 2018):

$$V_s = aD^b \tag{14}$$

$$V_c = \frac{d}{T} = \frac{d}{v/Q} = \frac{d}{(d \times RA)/Q} = \frac{dQ}{dRA} = \frac{Q}{RA} = overflowrate \tag{15}$$

$$TE = 100 \times \frac{V_s}{V_c} = 100 \times \frac{A_{wb}}{Q} \times V_s \tag{16}$$

     where TE is the trapping efficiency, $V_s$ the settling velocity (Eq.14) and $V_c$ the critical settling velocity. The constants a and

b were given a value of 710 and 1.57 respectively and the median grain size D was assumed $5 \times 10^{-6}$ m derived from the





research of Thonon et al. (2005). The settling velocity $V_s$ was therefore assumed at $3.38 \times 10^{-6}$ m/s. The $V_c$ (Eq.15) can be related to the overflow rate using the water depth $d$, retention time $T$, reservoir volume $v$, surface area of the water body $A_{wb}$ and the in or out flowing discharge $Q$ (Haan et al., 1994). The calculation of the discharge is described in Appendix A.

### 3.4   Model validation

As we intend to use RDSM for future projections of climate change and land cover dynamics, we choose not to calibrate the model. Yet, we validated the RDSM for a 30 year period ($1980 - 2009$) for the entire Amazon basin. This parameterization of this period includes historic land cover dynamics and reservoir construction where available. The first year was used as a spin-up period for the sediment dynamics, the hydrological model having a longer spin-up period to prime the groundwater module of PCRGLOB-WB which has a long memory. To validate the model, we compared model outcomes in terms of discharge

and sediment transportation to observations available in the Hybam dataset for seven gauging stations on the Amazon and its main tributaries. These stations were selected out of 14, as the Nazareth station is located very close to Tabatinga. The chosen stations have the longest records of sediment samples and comprehensively cover the entire basin, making them representative of the Amazon basin (Figure 1). In the Hybam dataset, the observed sediment concentration was typically sampled every ten days or three times a month at fixed positions near the middle of the river. However, the overall number of samples was sparse,

and not all stations are covered at all times, sometimes creating wide gaps in the coverage. For example, Tabatinga has one sample in 1995 and one in 1997, while there were no samples in 1996, 2008 and 2009. Moreover, there was a low number of samples for each year at Tabatinga ranging between 0 and 4, and at Manacapuru in 1995. On the other hand, near-continuous daily discharge values are available for all seven stations, except the Manacapuru station having some missing data for 2003 and 2004.

Discharge observations are fairly complete, so we calculated the monthly mean discharge from the modeled daily discharge values and compared these with the available monthly discharge observations from Hybam dataset. In our analysis, we first validated the discharge data since these observations are more complete, and the accuracy of simulated sediment transport is highly dependent on discharge. However, sediment is only sampled sporadically. To make use of the sparse observations and to facilitate the comparison between the observed and simulated values, the following steps were taken to calculate the observed

monthly and annual suspended sediment transport (Eq.17-19).

$$St_d = Si_d \times dis_d \tag{17}$$

$$St_{monthly} = \frac{N_{monthly}}{Ns_{monthly}} \times \sum Si_d \tag{18}$$

$$St_{yr} = \frac{N_{yr}}{Ns_{yr}} \times \sum Si_{monthly} \tag{19}$$





Where $St_d$: daily sediment transport [kg/day], $St_{monthly}$: monthly sediment transport [kg/month], $St_{yr}$: annual sediment

transport [kg/year] , $N_{monthly}$: number of days in the month, $Ns_{monthly}$: number of samples in the month, $N_{yr}$: number

of days of the year. $Ns_{yr}$: number of samples in the year. $si_d$:daily (ten days) sediment concentration [kg/day], $Si_{monthly}$:

monthly sediment concentration [kg/month].

The model performance was evaluated using Kling-Gupta efficiency equation KGE (Eq.20)and the relative value of Root

Mean Square Error equation RMSE (Eq.21, Eq. 22). The KGE is a model evaluation criterion presenting the contribution of

mean, variance and correlation on model performance, and it has been widely used in validating hydrological models in recent

years (Gupta et al., 2009; Liu et al., 2022). RMSE is one of the most commonly used measures for evaluating the quality of

model predictions which shows how far predictions deviate from measured values(Christie and Neill (2022)).

$$KGE = 1 - \sqrt{(CC-1)^2 + (\frac{c_d}{r_d} - 1)^2 + (\frac{c_m}{r_m} - 1)^2} \qquad (20)$$

where CC is Pearson coefficient value, $r_m$ is average observed values, $c_m$ is average simulated values, $r_d$ is standard devia-

tion of observed values , $c_d$ is standard deviation of simulation values

$$RMSE = \sqrt{\frac{1}{n} \sum_{i=1}^{n} (x_{obs,i} - x_{sim,i})^2} \qquad (21)$$

where $x_{obs}$ is observation values and $x_{sim}$ is simulation values.

$$abRMSE = \frac{sim}{obs} - 1 \qquad (22)$$

where sim is the simulation values and obs is the observation values.

Sediment production is spatially localized and the propagation of the sediment along the drainage system determines the

final sediment output from the Amazon Basin to the ocean, as estimated by earlier studies. To analyze the amount of erosion

delivered to the river, sediment production is calculated per catchment [tonne/year] and per catchment area [tonne/km$^2$/year],

Eq.23 and 24. To analyse how sediment operate across the Amazon basin, Equations 25, 24 are used.

$$S_{pro} = \sum A + \sum S_{del} \qquad (23)$$

$$S_{proarea} = \frac{S_{pro}}{\sum area} \qquad (24)$$

$$S_{in} = S_{pro} + S_{trin} \qquad (25)$$



$$S_{dep} = S_{in} - S_{trout} \qquad (26)$$

Where $S_{pro}$: annual sediment production in the catchment, A: annual soil loss (erosion) in the catchment[tonne/year], $S_{del}$: annual sediment delivered to the river in the catchment [tonne/year], $S_{proarea}$: sediment production per catchment area

[tonne/km²/year], area: cell area in the catchment [km²]. $S_{in}$: sediment inflow, $S_{trin}$: sediment transport from the upstream area into the catchment [tonne/year], $S_{dep}$: sediment deposition in the catchment [tonne/year], $S_{trout}$: annual sediment transport out of the catchment [tonne/year].

## 4 Results & Discussion

### 4.1 Sediment Production

The sediment loss constitutes an upper limit of the sediment production that can be delivered to the streams. The average sediment production map from $1980 - 2009$ shows that most of the sediment originates from the Andes (Figure 5), supporting the theory that $80 - 95\%$ of the sediment production originates there (Filizola Jr et al. (2011); Latrubesse et al. (2005); Meade et al. (1985); Meade (1994)). These large quantities of sediments are produced by erosion as a result of the high rainfall rate combined with the steep topography in this area (Martinelli et al. (1989a)). In addition, Figure 5 shows that appreciable

sediment production also occurs in the south of Brazil.

These results corroborate the findings of (Gomes et al., 2019), who found that agriculture activities associated with deforestation and inadequate management increased soil erosion in southern Brazil–Cerrado resulting in an increased annual rate of $10.4 - 12.0$ tonne/ha/yr between 2000 and 2012.

Further, RPSM simulated the spatial distribution of monthly sediment production in the Amazon basin between $1980 -$

2009. The average annual sediment production was assessed per catchment (Figure 1) and per catchment area (Table 1). The highest value of average sediment production was at the catchment of Tabatinga followed by Protovelho, Serrinha, Caracarai, Manacapuru, Fazenda vista Alegre, and Obidos Porto. A possible explanation for these results is that Tabatinga is closer to the Andes ( Figure 5), the major source of sediment.







**Figure 5.** Map of annual average sediment production $1980 - 2009$. Most of the sediment production originates from the Andes and from the south of the Brazil-Cerrado as it is shown in the figure

## 4.2 Discharge

The simulated annual average discharges were in good agreement with the measured values at seven stations: Obidos Porto, Manacapuru, Portovelho, Fazenda vista Alegre and Caracarai. At Tabatinga, there was an underestimation, while the annual simulated values at Sirrenha showed overestimations (Table 2). The monthly simulated and observed discharges 1980 - 2009 (Figure 6) show that the peak discharges had good agreement at all stations. The simulated interannual variation in discharge is well captured by the model according to the KGE and RMSE, especially downstream and on the Rio Madeira tributary.

Obidos Porto has KGE of 0.917 and RMSE of $6.28 \times 10^4$ m$^3$/s. In addition, Portovelho and Fazenda vista Alegre has KGE of 0.823 and 0.8 and RMSE of $6.11 \times 10^3$ and $1.16 \times 10^4$ m$^3$/s, respectively (Table 2). Furthermore, the seasonality is well captured when compared to the observations (Figure 7).However, there are biases that vary between stations. Stations such as



**Table 1.** The main results of annual sediment production per catchment and per catchment area

| Station | tonne/yr | tonne/km$^2$/yr |
|---|---|---|
| Tabatinga | $34.3 \times 10^8$ | 389.2 |
| Manacapuru | $0.71 \times 10^8$ | 5.444 |
| Obidos Porto | $0.04 \times 10^8$ | 0.547 |
| Portovelho | $13.2 \times 10^8$ | 135.0 |
| Fazenda vista Alegre | $0.06 \times 10^8$ | 2.018 |
| Serrinha | $0.26 \times 10^8$ | 9.097 |
| Caracarai | $0.01 \times 10^8$ | 7.452 |

**Table 2.** The main results of the simulated and observed annual average discharge and the model performance for monthly discharge (KGE and RMSE)

| Stations | sim.[a] m$^3$/s | obs.[b] m$^3$/s | KGE | RMSE m$^3$/s | abRMSE.[c] |
|---|---|---|---|---|---|
| Tabatinga | $2.62 \times 10^4$ | $3.60 \times 10^4$ | 0.603 | $1.12 \times 10^4$ | $-0.27$ |
| Manacapuru | $9.16 \times 10^4$ | $10.2 \times 10^4$ | 0.815 | $4.07 \times 10^4$ | $-0.10$ |
| Obidos Porto | $17.5 \times 10^4$ | $17.2 \times 10^4$ | 0.917 | $6.28 \times 10^4$ | 0.017 |
| Portovelho | $2.00 \times 10^4$ | $1.89 \times 10^4$ | 0.823 | $6.11 \times 10^3$ | 0.05 |
| Fazenda vista Alegre | $3.05 \times 10^4$ | $2.79 \times 10^4$ | 0.8 | $1.16 \times 10^4$ | 0.09 |
| Serrinha | $2.24 \times 10^4$ | $1.64 \times 10^4$ | 0.57 | $5.73 \times 10^3$ | 0.36 |
| Caracarai | $0.285 \times 10^4$ | $0.290 \times 10^4$ | 0.795 | $2.31 \times 10^3$ | $-0.02$ |

[a] simulations, [b] observations, [c] : absolute RMSE

Obidos Porto, Portovelho, Fazenda Vista Alegre and Caracarai exhibit less bias compared to the others (Figure 7). There were underestimations at the mainstream stations of Tabatinga (Figure 6d) and Manacapuru (Figure 6e), which may be related to
the low estimation of precipitation in the Hybam input data. Because of the lack of available rainfall gauges in the western region (Andean sub-basins) of the Amazon (Hoch et al., 2017). Elsewhere more uniform precipitation and denser observations could explain the lower discrepancies at Obidos Porto compared to Tabatinga and Manacapuru. Higher performance could also partly be explained by the complementary monsoon seasons in the eastern, lower part of the Amazon (Guimberteau et al., 2011). Another source that can influence the simulate discharge is the groundwater system which is often too slow to respond
in the model, which might partly explain the overestimation of the low flows. For example, in the Rio Negro tributary, there was an overestimation at Serrinha (Figure 6d), while the simulations for Caracarai (Figure 6f) matches very well with the observed values at high and low flow. The higher estimated Hybam precipitation in the north region of the basin might explain the overestimation at Serrinha. Overall, the agreement between simulations and observations indicates good performance of the hydrological model and its ability to represent discharge.

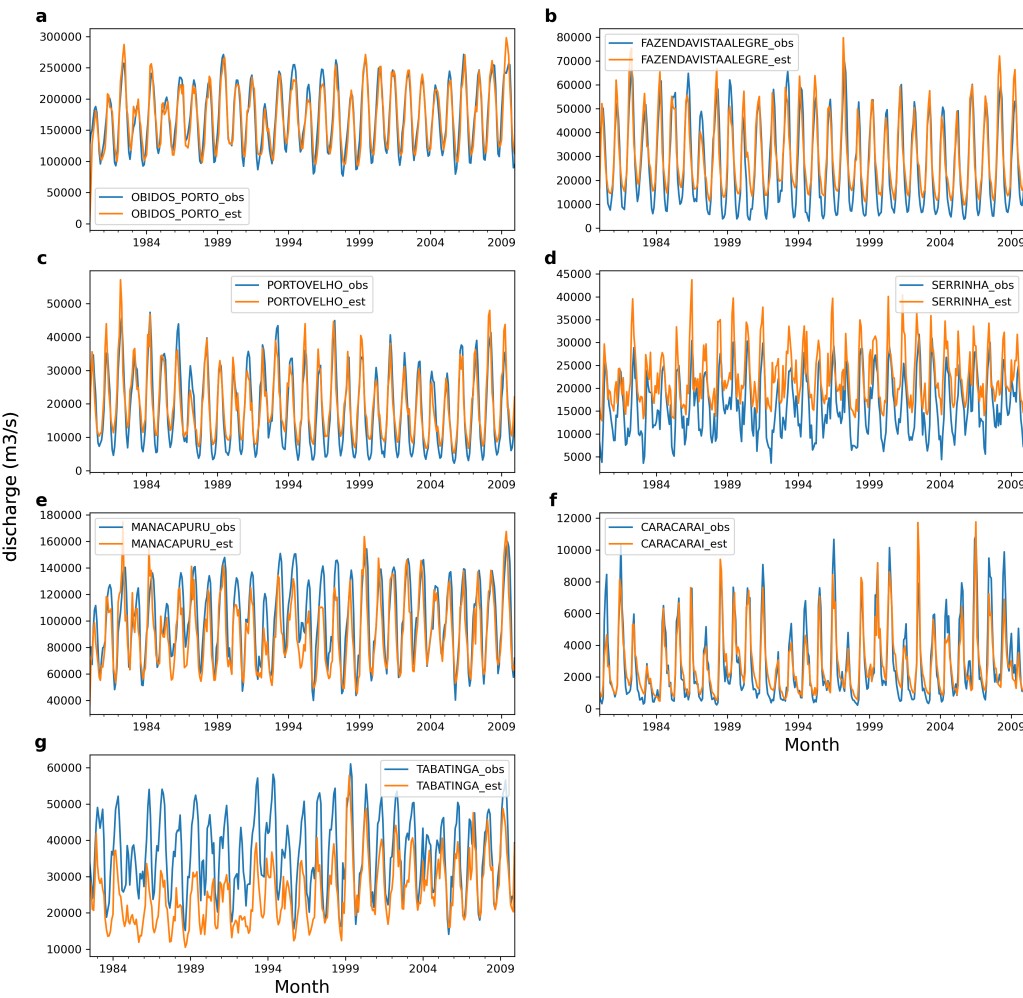

**Figure 6.** Simulated and observed monthly discharge at seven stations in the Amazon basin ($1980 - 2009$). The simulated average monthly discharge matches well at Obidos Porto (6a), Portovelho (6c), Fazenda Vista Alegre (6b) and Caracarai (6f). At Tabatinga (6g) and Manacapuru (6e), there was underestimations, while there was an overestimation at Serrinha (6d)

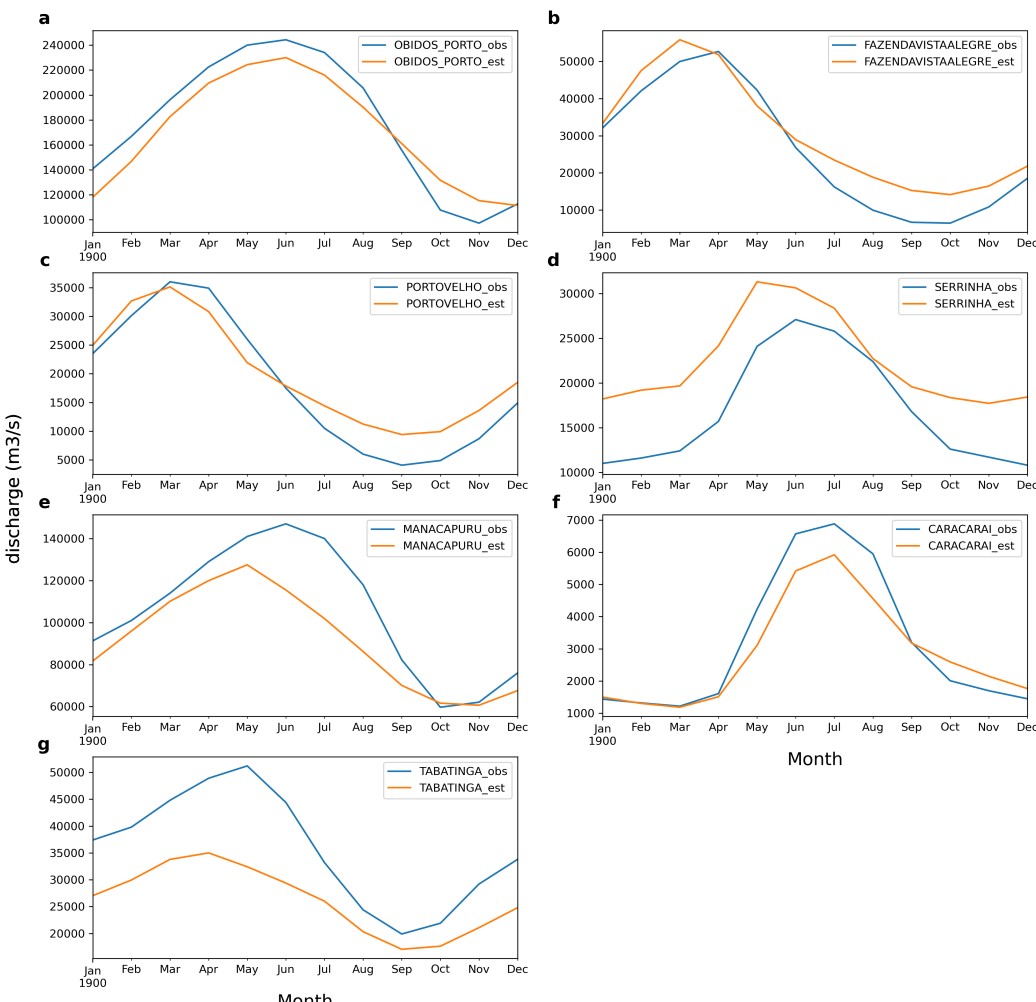

**Figure 7.** Seasonal discharge at seven stations compared to the observations over the period $1980 - 2009$. To calculate the seasonality, the monthly discharge was averaged for each month across all years. there are biases that vary between stations. Stations such as Obidos Porto (7a), Portovelho (7c), Fazenda Vista Alegre (7b) and Caracarai (7f) show less bias compared to the others

## 4.3 Sediment Transport

### 4.3.1 At station level

The simulated annual sediment transport showed agreement with the observations at Portovelho and Caracarai. Moreover, it shows an overestimation at Tabatinga, Manacapuru, Obidos Porto,Fazenda Vista Alegre and Serrinha (Table 3).





The monthly simulated and observed sediment during $1980 - 2009$ showed an agreement in timing of the peaks compared
to the observations. Obidos Porto (Figure 9a) and Portovelho (Figure 9c) provide more complete observational records than
the other stations, and their peaks are better represented (Table 4).

The seasonality is well captured by the model for all stations, even though there are differences in the mean variation between
observations and estimations. Tabatinga (Figure 9d), Manacapuru (Figure 9c), Obidos Porto (Figure 9a), Portovelho (Figure
9c) and Fazenda vista Alegra (Figure 9b) show one peak from winter to early spring. At Serrinha (Figure 9d) and Caracarai
(Figure 9f) stations, the peaks occur in July. These peaks are effectively simulated bearing in mind that the maximum sediment
transport in Rio Negro tributary occurs at the second half of the year. While the peak of sediment transport at mainstream of
the Amazon river occurs at the first half of the year.

The sediment transport is dependent on discharge and sediment delivered to the river in the model, accordingly, the process
dynamic is covered.

The impacts of the trapping efficiency of water bodies can be observed at Manacapuru and Obidos Porto, where the sediment
deposited at Ria Lake reduced the sediment transported to Manacapuru. Although sediment was also deposited in the Curuai
Lake and the reservoir before Obidos Porto, it was compensated with the sediment coming from the other main tributaries.

One possible reason for the overestimation at Tabatinga, Manacapuru, and Obidos Porto was the high amount of sediment
production at the cell level, which was added in the model to the incoming sediment from the upstream cells, and then passed
to the downstream cells.

The differences of the sediment between upstream and downstream part of the Amazon river indicates that the contribution
of the tributaries to the main channel is significant and the sediment transport is dependent on the transport capacity and the
contribution of the tributaries to the main stream. Moreover, the decline of the sediment indicates that the sediment transport is
limited to the transport capacity and the sediment trapped in the reservoirs.



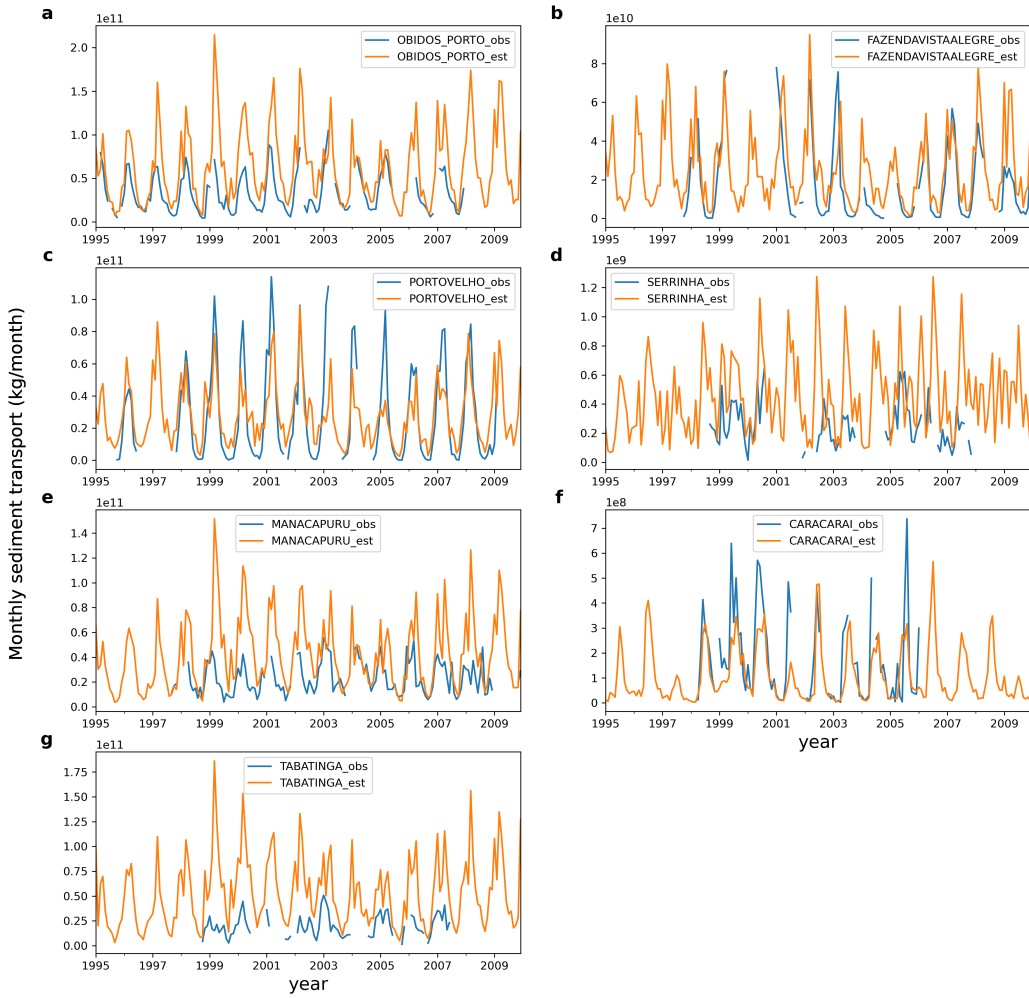

**Figure 8.** Simulated and observed monthly sediment transport at seven stations in the Amazon basin. The model estimations show good agreement in the peaks compared to the observation at all stations. The simulated monthly sediment transport showed agreement with the observations at Portovelho (8c) , Fazenda vista Alegre (8b) and Serrinha (8d). There was an underestimation at Caracarai (8f) and overestimations at Tabatinga (8g), Manacapuru (8e) and Obidos Porto(8 a).

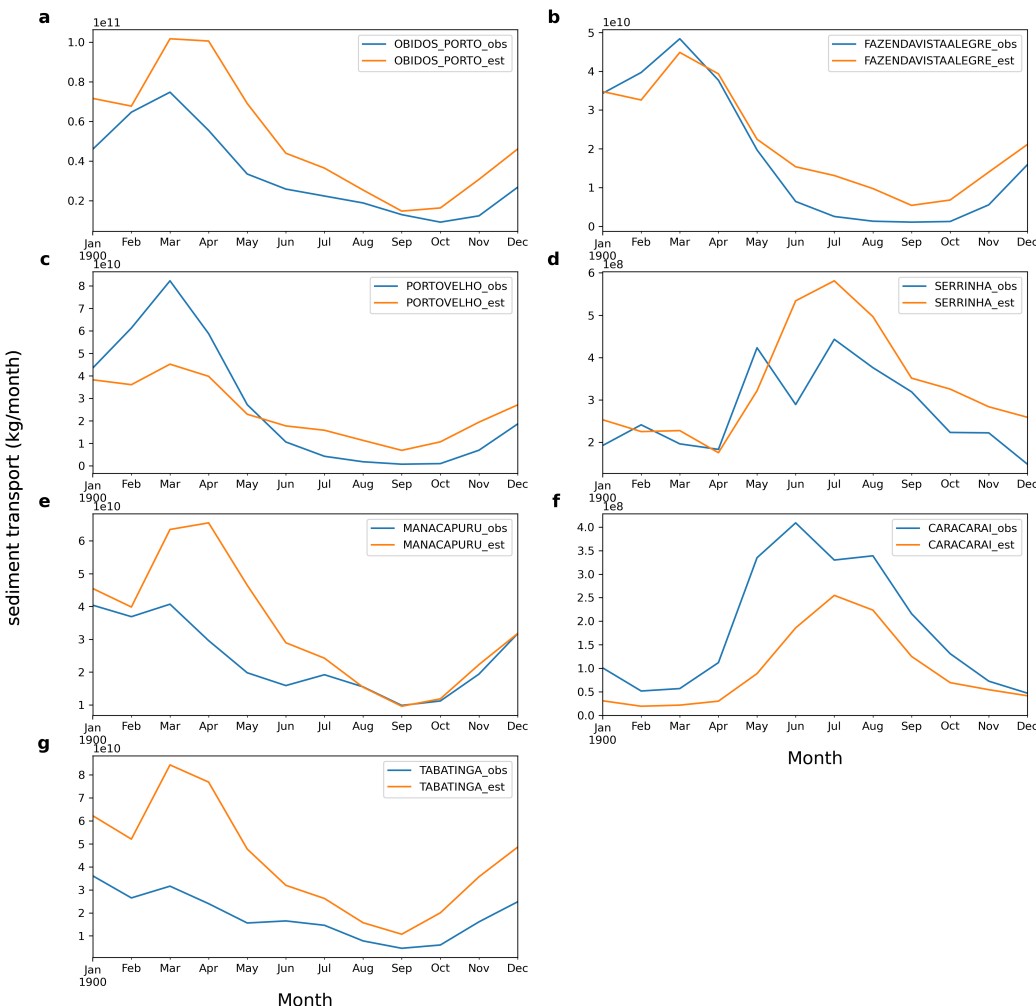

**Figure 9.** Seasonal sediment transport at seven stations compared to the observations over the period $1980-2009$. To calculate the seasonality, the monthly sediment transport was averaged for every month of all years

### 4.3.2 Basin analysis

The sediment production, transport (inflow and outflow) and deposition was calculated per catchment, with the total inflow (sediment production, sediment inflow) equal to the total outflow (sediment deposition, sediment outflow) per catchment (see section 3.5 in model validation ). Sediment deposition occurred in canals and lakes. The catchments of Tabatinga, Portovelho,





Serrinha and Caracarai (Figure 1) are dependent on internally produced sediment due to the absence of sediment inflow from
upstream basins. Figure 10 shows the percentage of the sediment inflow and outflow in each catchment. For example, the
highest amount of sediment production was in Tabatinga catchment due to the Andes (see 4.2 section). However, 43.6% of the
sediment is again deposited in the same catchment while only 6.4% is transported to other catchments downstream.

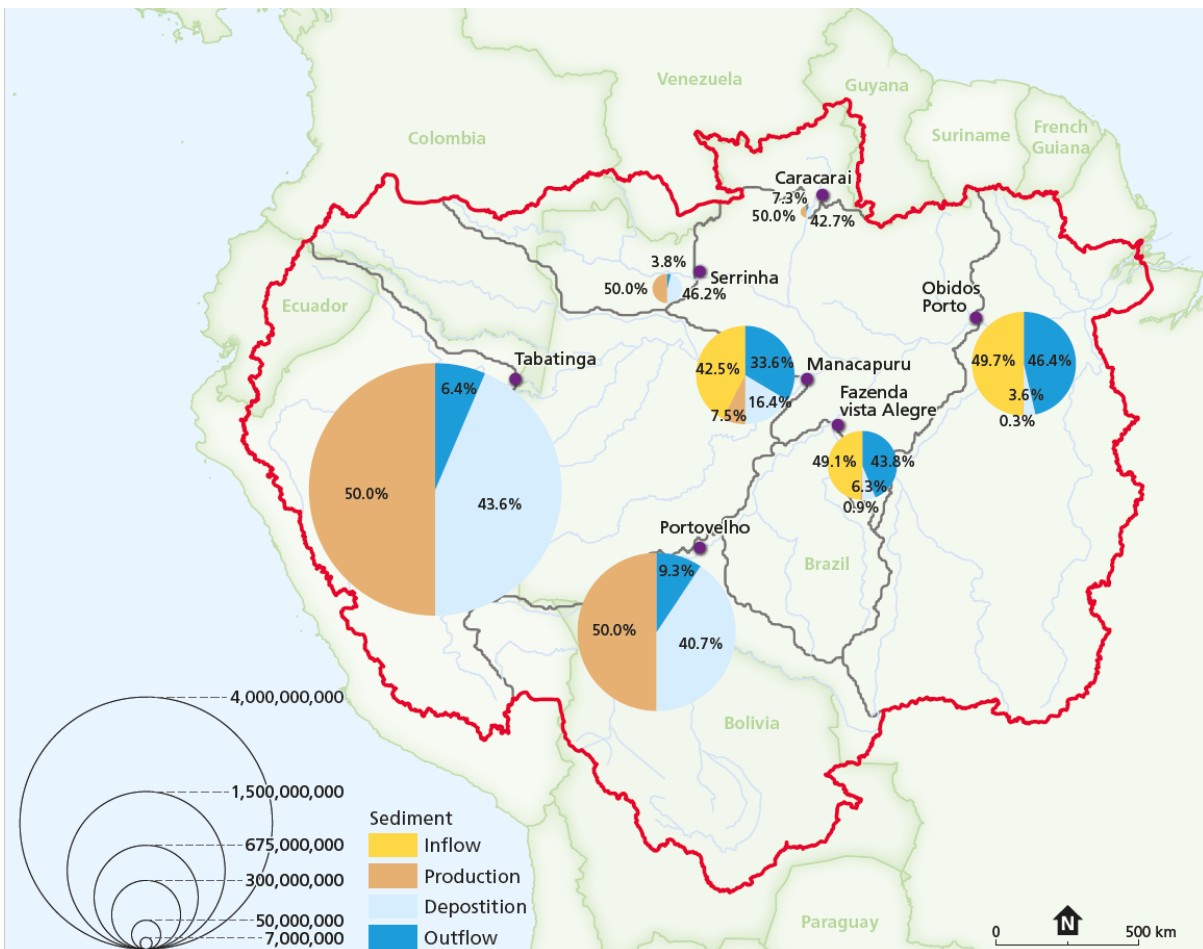

**Figure 10.** The figure shows the proportion of inflow (sediment production in the catchment, sediment inflow into the catchment) and the
outflow (sediment deposition, sediment outflow from the catchment) in the basin. The area of the pie was proportional to the total inflow. The
inflow was at the right side and colored in yellow and brown while the outflow was at the left side and colored in dark and light blue. The
inflow and outflow were equally divided 50% each side.Tabatinga and Portovelho, Serrihna and Caracarai have 50% of sediment production
due to the absence of upstream stations and therefore no sediment inflow

### 4.3.3   Previous studies

Except for the study by  (Hatono and Yoshimura, 2020), the sediment transport that RDSM simulates at the stations match
closely with the values reported in the literature (see Table4). The study of  (Hatono and Yoshimura, 2020) employed a global





sediment model in which erosion was based on slope, precipitation, and two coefficients. While their model described the total sediment transport as a result of suspended and bedload transport, the expected sediment transport is generally lower than that of RDSM and the other studies with the exception of the station of Serrinha. The other studies are observation based; (Wittmann et al., 2011) compiled suspended sediment transport data from various sources, and reported the minimum and

maximum values per station; (Lima et al., 2005) collected the suspended sediment concentration from each station in the Brazilian Hydrometric Basic Network and multiplied it with the discharge to obtain the sediment transport; (Filizola Jr and Guyot, 2009) again used data from the Brazilian Hydrometric Basic Network but applied three distinct methods to convert the sediment concentrations into the sediment transport. In their approach, the first method involved a specific linear relationship between suspended sediment transport and the sediment concentration,the second method relied solely on the discharge to

estimate sediment transport, and the third method accounted for both the discharge and the rate of change in water slope over time. If we evaluate our simulations in relation to the observation-based estimates, we observe not only the agreement in terms of the mean annual sediment transport at the stations but also that our estimate is robust and centred on the more likely values per station. Also, the added model complexity in our study is in agreement with the non-linear estimates by (Filizola Jr and Guyot, 2009) that introduces hysteresis by adding the rate of change in the water slope. While the bias at station level is small as

evidenced by the RMSE and the station comparison (see Table 3 and Table 4 ), the added dynamics of the model are not directly reflected in the skill of the model, as the KGE is low (KGE <= $0.414$ does imply no skill at all) (Table 3). However, the quality of the observations for the validation of the sediment transport is severely restricted in both this and existing literature study as measurements of the sediment concentration are sparse and may have low representatives because of their small volumes. Notwithstanding, the sediment transport modelled by RDSM behaves well in terms of its spatial patterns and probably temporal

dynamics, which is remarkable as the model is not calibrated.

### 4.3.4   Sediment transport to the ocean

The Figure (11) shows the temporal dynamics of sediment production and sediment transport from $1980 to 2009$. It illustrates the impacts of climate, land cover variations, and sediment deposition by water bodies on these processes.

From the simulated results, the Amazon river transports annually $5.96 \times 10^8 \mathrm{tonne/yr}$ to the ocean. This estimate aligns with

the findings of (Gibbs, 1967b) and (Filizola Jr and Guyot, 2009), (Oltman, 1968) and (Mouyen et al., 2018)and closely matches the simulations by (Fagundes et al., 2023) (Table 5). It is also within the same order of magnitude as estimates from (Pelletier, 2012). However, our model's estimate is higher than the values reported by (Hoch, 2014) (Table **??**). The overestimation in comparison to (Hoch, 2014) is due to underestimation of the land cover impacts by PCRGLOB-SET. By incorporating these impacts into our model (see section 3.1 in Methods and Data), we calculated a sediment production input of $11.1 \times 10^8 \mathrm{tonne/yr}$

compared to (Hoch, 2014) $37.0 \times 10^8 \mathrm{tonne/yr}$. Additionally, accounting for the trapping efficiency of reservoirs (see 3.3 section in Methods and Data) results in a sediment deposition of $5.10 \times 10^8 \mathrm{tonne/yr}$,leaving $5.97 \times 10^8 \mathrm{tonne/yr}$ to be transported. The inclusion of these impacts has refined our model, aligning it more closely with (Filizola Jr and Guyot, 2009) findings that his model transported $39.0 \times 10^8 \mathrm{tonne/yr}$ before including reservoirs, which adjusted the transport to $4.06 \times 10^8 \mathrm{tonne/yr}$.



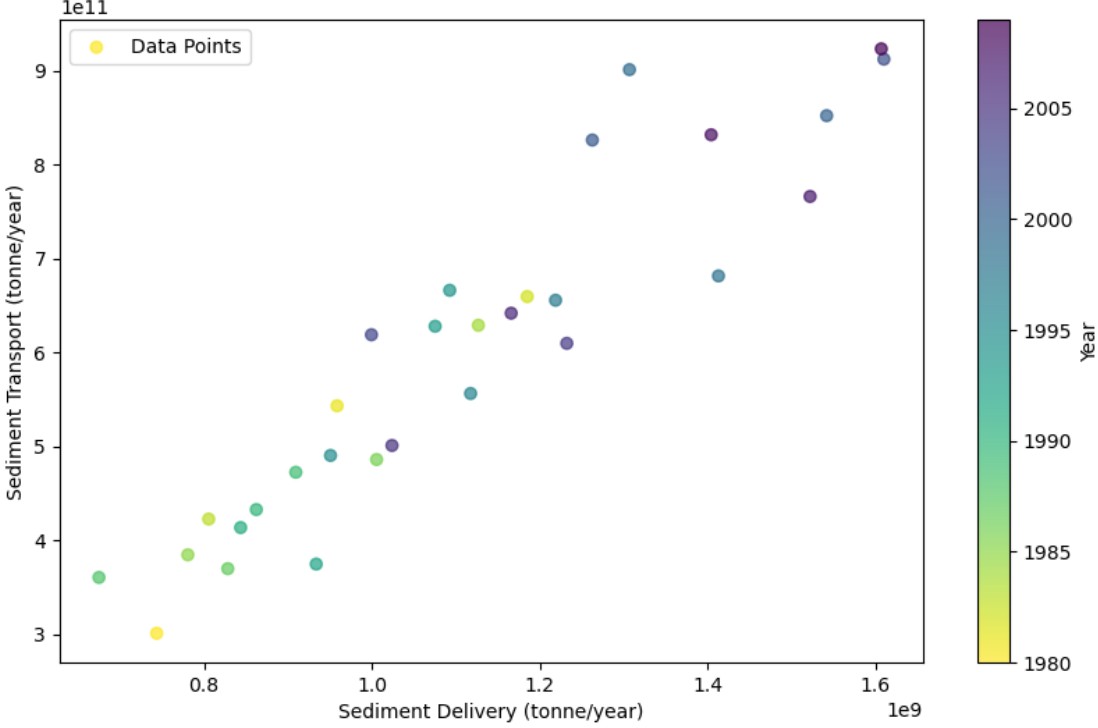

**Figure 11.** Sediment production and sediment transport $1980-2009$.The figure shows that the sediment transport increases with the increase of sediment production

**Table 3.** Observed and simulated annual average sediment transport and the model performance for monthly sediment transport (KGE and RMSE)

| Stations | sim.[a] tonne/yr | obs.[b] tonne/yr | N[c] | Period | KGE | RMSE | abRMSE[d] |
|---|---|---|---|---|---|---|---|
| Tabatinga | $5.12 \times 10^8$ | $2.24 \times 10^8$ | 220 | $10/95 - 10/07$ | $-1.7$ | $4.07 \times 10^{10}$ | 1.29 |
| Manacapuru | $4.05 \times 10^8$ | $2.90 \times 10^8$ | 348 | $3/95 - 12/09$ | $-0.54$ | $3.24 \times 10^{10}$ | 0.39 |
| Obidos Porto | $6.46 \times 10^8$ | $4.03 \times 10^8$ | 393 | $3/95 - 12/09$ | $-0.33$ | $4.07 \times 10^{10}$ | 0.60 |
| Portovelho | $2.91 \times 10^8$ | $3.17 \times 10^8$ | 393 | $6/95 - 12/09$ | 0.49 | $1.68 \times 10^{10}$ | $-0.08$ |
| Fazenda vista Alegre | $2.60 \times 10^8$ | $2.14 \times 10^8$ | 307 | $10/97 - 12/09$ | 0.42 | $1.68 \times 10^{10}$ | 0.21 |
| Serrinha | $0.04 \times 10^8$ | $0.027 \times 10^8$ | 237 | $7/96 - 11/08$ | $-0.14$ | $3.55 \times 10^8$ | 0.48 |
| Caracarai | $0.01 \times 10^8$ | $0.011 \times 10^8$ | 347 | $12/96 - 12/09$ | 0.28 | $1.29 \times 10^8$ | $-0.09$ |

[a] simulations,[b] observations, [c]: number of samples, [d] :absolute RMSE





**Table 4.** Simulated annual average sediment transport of RDSM at stations compared to the simulations of other studies tonne/yr

| Stations | RDSM | literature [a] | literature [b] | literature [c] | literature [d] |
|---|---|---|---|---|---|
| Manacapuru | $4.05 \times 10^8$ | $4.03 - 7.43 \times 10^8$ | $4.52 \times 10^8$ | $4.02, 3.04, 4.01 \times 10^8$ | $2.3 \times 10^8$ |
| Obidos Porto | $6.46 \times 10^8$ | $5.51 - 13.2 \times 10^8$ | $5.67 \times 10^8$ | $5.55, 4.53, 6.14 \times 10^8$ | $4.01 \times 10^8$ |
| Portovelho | $2.91 \times 10^8$ | $2.3 \times 10^8$ | $2.43 \times 10^8$ | $2.77, 2.55, 262 \times 10^8$ | $0.9 \times 10^8$ |
| Fazenda vista Alegre | $2.60 \times 10^8$ | $1.51 - 7.15 \times 10^8$ | $2.38 \times 10^8$ | $2.44, 2.01, 2.43 \times 10^8$ | |
| Serrinha | $0.04 \times 10^8$ | | | $0.038, 0.033, 0.040 \times 10^8$ | $0.3 \times 10^8$ |
| Caracarai | $0.01 \times 10^8$ | | | $0.029, 0.027, 0.029 \times 10^8$ | |

[a] Wittmann et al. (2011), [b] Lima et al. (2005), [c] Filizola Jr and Guyot (2009), [d] Hatono and Yoshimura (2020)

**Table 5.** Simulated annual average sediment transport to the ocean $\mathrm{tonne/yr}$ of RDSM and other studies

| Annual average sediment transport to the ocean | |
|---|---|
| RDSM simulations | $5.96 \times 10^8$ |
| (Gibbs, 1967a) | $5.00 \times 10^8$ |
| (Oltman, 1968) | $6.00 \times 10^8$ |
| (Filizola Jr and Guyot, 2009) | $5.50 \times 10^8 - 10.3 \times 10^8$ |
| (Mouyen et al., 2018) | $7.78 \times 10^8$ |
| (Fagundes et al., 2023) | $4.06 \times 10^8$ |
| (Fagundes et al., 2021) | $39.0 \times 10^8$ |
| (Hoch, 2014) | $37.0 \times 10^8$ |
| (Pelletier, 2012) | $9.50 \times 10^8$ |

## 5 Conclusions

This paper introduces the River Discharge and Sediment model RDSM which contains reservoirs and their trapping efficiency and accounts for the impacts of land use in the Amazon. We applied the model at a spatial resolution of $5$ arc minutes over the period $1980 - 2009$. We validated it in terms of discharge and sediment transport using the Hybam database for seven gauging stations on the mainstream Amazon and its tributaries. We also compared the sediment transported to the ocean of our model to simulations from previous studies. Our model covers different spatial scales and links soil loss at the hillslope scale to sediment

entrapment and uptake along the river to the transport to the ocean. Information on each of these aspects and their connections are important to assess the effects of erosion for farming, of sediment transport and fragmentation for environmental purposes, and for the stability of the Amazon estuary and coasts.

The analyses of the sediment in the basin shows that the catchment of Tabatinga had the highest sediment production followed by the catchments of Portovelho, Serrinha and Caracarai.





Our analysis shows that the annual and monthly simulated discharge values agreed at most of the stations with reported values, with KGE values between $0.57$ and $0.92$. Further, the annual values for sediment transport shows agreement with the simulated values in most of the stations. Monthly and annual modelled sediment transport overestimated some stations compared to the Hybam observations, with KGE values between $-1.7$ and $0.49$.

       RDSM computes sediment transport to the ocean at $5.96 \times 10^8$ tonne/yr. It agrees with field measurements and has small
differences with previous studies due to the trapping efficiency impacts. The RDSM effectively represents the patterns of monthly and annual variations of discharge and sediment transport at $5$ arc minutes resolution in the Amazon basin and to the ocean.

       Our study has some limitations; a major problem is data availability. This concerns the underestimation in the precipitation as an input to the model that affects the simulated discharge and consequently the sediment transport. It also concerns the
observational data on sediment concentration, which was used to validate the model. The model also ignores some processes that are hard to model and parameterize. This concerns flood plain deposition and bank erosion. Their inclusion will make the model more complete but also more complex and sensitive to errors in any possible parameterization. It is not directly evident that their inclusion will automatically increase model performance. Other model improvements are a higher temporal resolution of the sediment delivery to the streams and a subdivision of the erosion processes adding to the sediment delivery in
the Andes, which is the biggest source of sediment for the Amazon but which in the current form of the model is represented in a rather simple form by the RUSLE.

       Still, our model and analysis contribute to our understanding of sediment transport dynamics and the impact of reservoirs on sediment transport in the Amazon basin. The model further facilitates the future estimation of sedimentation impact in reservoirs incorporating water resource management and can thus contribute to a better understanding of the complexity of the
Amazon basin, which is a key region in the biogeochemical and hydrological cycles of the Earth, and provide indispensable information to manage its resources more sustainably.

**Appendix A**

**A1    River Sediment Production Model  RSPM**

**A1.1    Slope length and steepness $LS$**

The slope length L is constant, reflecting the impact of slope length on erosion: the longer the slope length, the more erosion occurs. It defined as "the horizontal distance from the origin of overland flow to the point where either the slope gradient decreases enough that deposition begins or runoff becomes concentrated n a defined channel" (Wischmeier and Smith, 1978). Hoch (2014) used the following equation to determine L (Renard (1997)):

$$L = \left( \frac{l_c}{22.13} \right)^{\gamma} \tag{A1}$$





where $l_c$ is the contribution of slope length in horizontal projection m, $\gamma$ is a variable slope length exponent, and 22.13 is the RUSLE unit plot length in meters. Every grid cell has been assigned a value for $\gamma$ depending on the slope angle. While assigning a value to $\gamma$ is quite a straightforward operation, the determination of the $l_c$ is not. Accordingly, it was derived from PCRGLOB-WB as 10 arc minutes, and then resampled to 5 arc minutes for use in RSPM (Table A1).

**Table A1.** Slope length exponent as a function of slope angle (%)

| slope angle | 00.5 | 0.51.0 | 1.03.4 | 3.45 |
|---|---|---|---|---|
| Exponent $\gamma$ | 0.15 | 0.2 | 0.3 | 0.4 |

By incorporating the slope steepness factor S, the influence of slope on erosion is reflected , the steeper the slope, the more erosion can occur. Slope steepness has more influence on soil loss than slope length (Renard, 1997). The equation for calculating S developed by Nearing (1997) was used by Hoch (2014) for the slope steepness computation due to its advantage in minimizing the computation time as well as being suitable to slope angle up to or more than 22% (Nearing, 1997). S is calculated as:

$$S = -1.5 + \frac{17}{1 + e^{2.3 - 6.1\sin(\theta)}} \tag{A1}$$

where $\theta$ is slope angle in degrees.

### A0.1 Rainfall–runoff erosivity $R$

Rainfall–runoff erosivity describes the potential of rainfall and resulting runoff to erode soil. The more intense the rainfall, the more soil can potentially be eroded through the impact of falling rain drops. (Renard and Freimund, 1994) derived the subsequent relation between the $R$–factor and annual precipitation in millimetre, based on 132 stations in the US. The annual amount of precipitation $P_{yr}$ is determined by aggregating monthly values of precipitation in millimetre $P_m$.

$$P_{yr} = \sum_{i=1}^{12} P_{m,i} \tag{A2}$$

Depending on the total annual precipitation in each grid cell, either a linear or quadratic function is applied to calculate the annual $R_{yr}$ due to the good correlation shown between these two parameters in a previous study (Yu and J. Rosewell, 1996)

$$R_{yr} = 0.04830 \times P_{yr}^{1.1610} \qquad\qquad \text{if } P_{yr} \leq 850 \text{ mm/yr} \tag{A3}$$

$$R_{yr} = 587.8 - 1.219 P_{yr} + 0.00415 P_{yr}^2 \qquad\qquad \text{if } P_{yr} > 850 \text{ mm/yr} \tag{A4}$$

In order to calculate the monthly $R_m$, the approach of Hoch (2014) was used, which is based on the fact that annual precipitation is the sum of monthly precipitation. The monthly percentage distribution (Part,i) for each grid cell and every month (i = 1 . . . 12) over the year is calculated by dividing the monthly precipitation to the annual precipitation:





**Table B1.** Land cover types and Interception (Int) values used in the model

| nu | land cover type | Int values | ref. |
|----|-----------------|------------|------|
| 1 | Barren sparsely vegetated | 0.0 | Webster (2005) |
| 2 | Closed shrublands | 0.27 | Venkatraman and Ashwath (2016) |
| 3 | Croplands | 0.28 | Webster (2005) |
| 4 | Deciduous broadleaf forest | 0.25 | Webster (2005) |
| 5 | Deciduous needleleaf forest | 0.35 | Webster (2005) |
| 6 | Evergreen broadleaf forest | 0.25 | Webster (2005) |
| 7 | Evergreen needleleaf forest | 0.35 | Webster (2005) |
| 8 | Grasslands | 0.25 | Webster (2005) |
| 9 | Mixed forest | 0.25 | Webster (2005) |
| 10 | Open shrublands | 0.27 | Venkatraman and Ashwath (2016) |
| 11 | Permanent wetlands | 0.25 | Webster (2005) |
| 12 | Savannas | 0.25 | Webster (2005) |
| 13 | Snow and ice | 0.0 | |
| 14 | Urban and builtup | 0.0 | |
| 15 | Water | 0.0 | |
| 16 | Woody savannas | 0.13 | Webster (2005) |

$$Part_i = \frac{P_m}{P_{yr}} \quad 0 \leq Part_i \leq 1 \tag{A5}$$

$$\tag{A6}$$

Then, the $R_m$ is calculated by multiplying the monthly fraction distribution Part,i with the annual $R_{yr}$:

$$R_m = Part_i \times R_{yr} \tag{A7}$$

## B1   Soil erodibility $K$

The $K$–factor represents the soil response (average long–term and profile) to the erosive power of rainfall which in turn triggers
soil detachment and transport due to surface shear and raindrop impact (Hoch, 2014).

The annual $K_{yr}$ (Mg ha h/ha MJ mm) was simulated by using the modified soil erodibility computation of the USLE as
presented by Torri et al. (1997). The reason for selecting this approach is its good performance based on Yang et al. (2003) as
well as the availability of the input data used in PCRGLOB-SET including fractions of sand ($f_{sand}$), silt ($f_{silt}$) and clay ($f_{clay}$)





**Table C1.** $K_{ratio}$'s and texture classes

| Texture class | Definition | $K_{ratio}$ |
|---|---|---|
| Coarse | $< 18\%$ clay and $> 65\%$ sand | 4.5 |
| Medium | $\leq 35\%$ clay and $\leq 65\%$ sand | 1.44 |
| Fine | $> 35\%$ clay | 1.17 |

up to 30 cm soil depth per grid cell, the amount of organic material OM %, and the soil texture classes. Only three topsoil

classes are selected due to their availability in the database at the scale of the map 1:5 million (fine, medium and coarse). Due

to the division by $f_{clay}$ in Eq. B2, it may happen that it is not defined for some grid cells where $f_{clay} = 0$. In this case, the

minimum value of the entire map has been assigned to the specific grid cell. Each dataset is then resampled to 10 arc minutes

and 5 arc minutes resolution for RSPM. The calculation of $K_{yr}$ is as follows:

$$D_g = -3.5f_{sand} - 2.0f_{silt} - 0.5f_{clay} \tag{B1}$$

$$\text{Expo} = -0.0021\frac{\text{OM}}{f_{clay}} - 0.00037(\frac{\text{OM}}{f_{clay}})^2 - 4.02f_{clay} + 1.72f_{clay}^2 \tag{B2}$$

$$K_{yr} = 0.0293(0.65 - D_g + 0.24D_g^2)e^{\text{Expo}} \tag{B3}$$

To calculate the $K_m$, the same equation used by Hoch (2014) was applied. It includes the seasonality ratio ($K_{ratio}$) which

is based on the soil texture, the time fraction of a month of snowmelt occurrence ($MT_{fr,i}$), and the soil erodibility that applies

on average over the year ($K_{yr}$). $K_{ratio}$ is the ratio of the seasonal to annual $K$, representing soil erodibility linked to the

seasonality values, derived based on the fractions of the selected soil textural classes (TableC1). For grid cells with no class

"non" has been defined and its $K_{ratio}$ was set to the median of $K_{ratio}$ values (Table C1).

$$K_m = K_{ratio}MT_{fr,i}K_{yr} + (1 - MT_{fr,i})K_{yr} \tag{B4}$$

The $MT_{fr,i}$ is determined by applying Eq.B5 used in PCRGLOB-SET

$$MT_{fr,i} = \frac{T_{max} - \text{SMT}}{T_{max} - T_{min}} \quad 0 \leq MT_{fr,i} \leq 1 \tag{B5}$$

## C1    Manning's surface roughness coefficient n

The Manning's roughness coefficient introduces the experienced resistance by the overland flow in its path to the channel. $n$ is a

function of irregularities of the surface ($n_1$), obstructions ($n_2$), vegetation cover ($v_3$) and the specific height and characteristics

of vegetation ($n_4$).





**Table D1.** Slope angle classes and associated surface irregularities roughness values ($n_1$)

| s% | $\leqslant 1$ | $> 1- \leqslant 2$ | $> 2- \leqslant 3$ | $> 3- \leqslant 4$ | $> 4- \leqslant 5$ | $\leqslant 5$ |
|---|---|---|---|---|---|---|
| $n_1$ | 0.001 | 0.014 | 0.02 | 0.026 | 0.039 | 0.05 |

$$n_m = n_1 + n_2 + v_{3,m} \times n_4 \tag{C1}$$

Due to a lack of data, $n_1$ is simulated by slope (s) because the steeper the slope the more irregularities can be assumed. $n_1$ was computed as a weighed average of slope angle percentage per cell and the corresponding value due to surface irregularities (Table D1).

The value of the $n_2$ factor is set to $0.049$ which is associated with obstruction of $15\%$ of the area according to (Hoch, 2014). In the calculations of the dynamic vegetation cover factor, $v_3$, monthly variations in Manning's surface roughness coefficient

stem all from variations in vegetation cover. Monthly NDVI maps for the Amazon basin were used. NDVI values range between $-1$ and $+1$. A value of $0$ is assumed as a minimum value, representing no vegetation cover in the grid cell, and $1$ as a maximum value, representing full vegetation cover in the grid cell. When NDVI $\geqslant 0$, then $v_3$ = NDVI. When NDVI $< 0$, then $v_3 = 0$. Last, the specific height and characteristics of vegetation need to be considered represented in $n_4$. To that end, data about land cover types from PCRGLOB-WB have been associated with specific roughness values. The different land cover types as used in

PCRGLOB-WB (Sutanudjaja et al., 2018) are listed (TableE1). The values range between $0$ and $1$ and represent the fractional coverage of the grid cell with a specific land cover type (Hoch (2014)).

### E1    River Sediment Transportation Model RSTM

To calculate the discharge, the kinematic wave approximation of the Saint–Venant equation is used. This equation consists of continuity and momentum equations:

$$\frac{\delta Q}{\delta x} + \frac{\delta A}{\delta t} = q \qquad \text{(continuity)} \tag{E1}$$

$$A = \alpha Q^\beta \qquad \text{(momentum)} \tag{E2}$$

Where Q is discharge through channel [m/s], x is the length of the channel [m], A is the wet channel cross–section [$m^2$], t is the elapsed time [s], q is the inflow per length of channel [$m^2/s$] and $\alpha$, $\beta$ are empirical constants. The two aforementioned equations can be combined into one equation:

$$\frac{\delta Q}{\delta x} + \alpha\beta Q^{(\beta-1)} \times \frac{\delta Q}{\delta t} = q \tag{E3}$$





**Table E1.** Available data on land cover and associated roughness coefficients $n_4$

| nu | land cover type | $n_4$ | ref. |
|----|-----------------|-------|------|
| 1 | Barren sparsely vegetated | 0.0113 | Arnoldus (1977) |
| 2 | Closed shrublands | 0.40 | Arnoldus (1977) |
| 3 | Croplands | 0.04 | Diodato (2004) |
| 4 | Deciduous broadleaf forest | 0.36 | Arnoldus (1977) |
| 5 | Deciduous needleleaf forest | 0.36 | Arnoldus (1977) |
| 6 | Evergreen broadleaf forest | 0.32 | Arnoldus (1977) |
| 7 | Evergreen needleleaf forest | 0.32 | Arnoldus (1977) |
| 8 | Grasslands | 0.368 | Arnoldus (1977) |
| 9 | Mixed forest | 0.40 | Arnoldus (1977) |
| 10 | Open shrublands | 0.40 | Arnoldus (1977) |
| 11 | Permanent wetlands | 0.086 | Arnoldus (1977) |
| 12 | Savannas | 0.368 | Dee et al. (2011) |
| 13 | Snow and ice | 0.001 | Hoch (2014) |
| 14 | Urban and builtup | 0.015 | Diodato (2004) |
| 15 | Water | 0.023 | Diodato (2004) |
| 16 | Woody savannas | 0.368 | Hoch (2014) |

The Manning equation (Chow et al. (1988)) was used to obtain $\alpha$ and $\beta$:

$$Q = \frac{R^{\frac{2}{3}} \times \sqrt{S}}{n \times A} \tag{E4}$$

$$R_h = \frac{A}{P} \tag{E5}$$

Where S [m/m] is the slope gradient, n is the Manning's roughness coefficient and $R_h$[m], is the hydraulic radius, A [m$^2$]

is the surface area, P [m] is the wetted perimeter. For this research n was set to $0.04$ at the channel and $0.1$ for the floodplain. Using A and P we can rewrite the equation:

$$A = \left(\frac{np^{\frac{2}{3}}}{\sqrt{S}}\right)^{\frac{3}{5}} \times Q^{\frac{3}{5}} \tag{E6}$$

$$\alpha = \left(\frac{np^{\frac{2}{3}}}{\sqrt{S}}\right)^{\frac{3}{5}} \tag{E7}$$

$$\beta = \frac{3}{5} \tag{E8}$$



**Table F1.** Model input data used for both sub models RSPM and RSTM

| Data | Source | Original resolution | References |
|---|---|---|---|
| Slope angles map (Appendix $A$) | PCRGLOB-WB | 10 arc minutes | Sutanudjaja (2018) |
| Slope length (Appendix $A$) | PCRGLOB-WB | 10 arc minutes | Sutanudjaja (2018) |
| Precipitation | Hybam [a] | 1° | ORE-HYBAM) (2018) |
| Soil properties | Harmonized World Soil Database | | FAO et al. (2012) |
| Temperature | WorldClim project 1970-2000 | 5 arc minutes | worldclim.org (2022) |
| NDVI | LP DAAC[b] | 10 arc minutes | |
| University (2023) | | | |
| Cover fraction maps | PCRGLOB-WB | 10 arc minutes | Sutanudjaja (2018) |
| Dams and reservoirs | HydroLAKES [c] | | Messager et al. (2016) |
| Channel width | PCRGLOB-WB | | Sutanudjaja (2018) |
| Channel length | PCRGLOB-WB | | Sutanudjaja (2018) |

[a] Hydrology and Geochemistry of the Amazon basin Hybam project database $(1980 - 2009)$, [b] The Land Processes Distributed Active Archive Center, located at the U.S. Geological Survey USGS Earth Resources Observation and Science EROS Center lpdaac.usgs.gov.These maps were modified and converted into netCDF file format at the Integrated Climate Data Center University of Hamburg,Germany, [c] globaldamwatch.org/grand

Due to its relationship with the water level, the wetted perimeter is calculated using the discharge in previous time step. The new discharge is calculated using the internal function in PCRaster that takes into consideration the discharge in previous time step, coefficients $\alpha$ and $\beta$ and lateral inflow q that passes over the local drainage direction LDD to the downstream cells. The lateral inflow is calculated from the direct inputs to the fresh water surface $I_w$ and the total drainage from the land surface at the end of the time step, the new stage h is calculated from the calculated discharge and the new stage is used to estimate the

wetted perimeter P.

$$q = \frac{Q_{total}}{\Delta L} = \frac{A_{cell}}{\Delta L}([1 - F_{racwater}] \sum [Q_i + F_{racwater} \times I_w] \tag{E9}$$

where q is the lateral inflow to the channel $[\mathrm{m^3/s}]$ , $F_{racwater}$ is the fraction of open water surface within a cell, $\Delta L$ is the length of the channel m, $I_w$ is the direct input to the freshwater surface $[\mathrm{m/d}]$ and $Q_i$ is the interflow per m slope width $[\mathrm{m^2/s}]$. In reservoirs and lakes, the water level was assumed to be static over time. In the flood plain, it is assumed that there

is no sediment deposition. In addition, the floodplain velocity $V_{fp}$ is equal to the velocity of the channel $V_c$. Accordingly, the flood plain deposition and bank erosion were ignored in the model. However, the floodplain is connected to the channel.





**Figure F1.** land cover types used in RDSM per sub basin $1970 - 2010$





**Table G1.** Stations Characteristics

| Station | Latitude | Longitude |
| --- | --- | --- |
| Tabatinga | -4.253 | -69.9333 |
| Manacapuru | -3.3106 | -60.6094 |
| Obidos Porto | -1.9472 | -55.5111 |
| Portovelho | -8.7483 | -63.9169 |
| Fazenda Vista Alegre | -4.8972 | -60.0253 |
| Serrinha | -0.4817 | -64.8272 |
| Caracarai | 1.8214 | -61.1236 |

*Author contributions.* SN, SdJ, RvB designed the initial study and methodology. SN, RvB and JH contributed to the modelling in this study. SN executed most of the data collection, modelling, analysis and validation with support of FD, JH and RvB. SN led the writing of the manuscript and all authors contributed significantly the writing and editing including GS

*Competing interests.* We have no competing interest

*Acknowledgements.* This project was financially supported by two research grants of NWO (Netherlands Research Organization: 1) Modelling the changing sediment yield of the Amazon for the next 100 years and the impact on the Surinamese coast (NWO VidW.1154.18.0147635); 2) Mangroves and Mud MangoMud: Monitoring and Modelling Coastal Dynamics in Suriname to Mitigate the Effects of Climate Change - Addressing Sustainable Development Goals 13: Climate Action and 4: Quality Education.NWO-WOTRO $W$07.303.106.





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
