# Peer review of "Discharge and sediment fluxes along the Amazon river: RDSM model concepts and validation"

_Hydrology and Earth System Sciences, 2024_

## Referee Comment (RC1)

**Title: Discharge and sediment fluxes along the Amazon river: RDSM model concepts and validation.**

The authors developed a new model for estimating sediment transport in large basins, having as a study case the Amazon basin. The new model is based on the large-scale hydrological model PCRGLOB-WB 2 and includes two modules: River Sediment Production Model RSPM and River Sediment Transport Model RSTM. The model was validated for a 30 year period (1980–2009) using suspended sediment data from HYBAM database.

The manuscript is well-written and presents several important previous studies. The authors mentioned that "Unfortunately, sediment production and transport modelling for the Amazon basin is a challenging task due to the complex nature of the soil erosion and sediment transport process.". In this way, and considering the introduction of the manuscript, the main aim of the study seems to be to overcome the problem of sediment transport estimations in the Amazon basin, with a more complex model that accounts for relevant processes. The methods are clear and well organized. The results are presented properly and the conclusion matches with the results.

MAJOR COMMENTS

In fact, improving the model and providing better estimates for this region is a very important and necessary task. However, the manuscript has a lot of important issues that require deeper reflection. After reading the entire work, I believe that there are many conceptual errors that did not achieve the objective of the work. Before I start my comments on this, I would like to reinforce that I carefully look the previous studies presented by the authors as well as the input data and modeling structure.

The first issue that caught my attention was the last paragraph of the introduction. The model is proposed to simulate changes, but the period of simulation ends in 2009. After that, I understood that your decision was based on the availability of the data (precipitation from Hybam). But if you need to change precipitation data in the future because Hybam database is limited, how do you can ensure the same performance of the model? More important than that, HYBAM protocols only allow to achieve total surface suspended solids (organic and inorganic), which is not consistent with the recommended protocol to get cross-sectional suspended sediment (inorganic) mass concentration (see ISO 4363 (2002) protocol). The profile of suspended sediment concentration shows greater values according to the increase of depth. I was looking for some papers about that in the Amazon basin, and Bouchez et al. (2011 - Prediction of depth-integrated fluxes of suspended sediment in the Amazon River: particle aggregation as a complicating factor) showed profiles of suspended particulate matter for several locations of the Amazon basin. At Óbidos gauge station, they showed values ~50-100 mg/L near the water surface and 300-600 mg/L near the river bed. This result shows that surface samples can be underestimated by around six times the traditional methods. In this sense, the validation process is compromised since the "truth" is not the real truth. Another serious flaw was assuming that the value measured on a single day is representative of an entire month. How can I assume that measuring 10% of the year (for example) can be representative of the whole year? The temporal variability is so high when we are talking about sediment concentration.

The second issue is concerning to represent relevant processes. You used the kinematic wave to route river discharge and neglected suspended sediment deposition in floodplains. Amazon basin has large flat areas. Floodplains and backwater effects are very important processes in this basin and should not be neglected in a work that has as one of the main goals to represent relevant processes.

There are some articles about this issue. Do not represent these phenomena will directly affect the results to be obtained. Both discharge and sediment load will have higher peaks and the timing when these peaks occur will be wrong too (which could be seen in your results). I realized that in the dry period, the model RDSM was overestimating the observed results. What evapotranspiration method and which data did you use to compute it?

The third issue is about the transport capacity for suspended sediment. At first, I thought it was a great idea to represent the erosion/deposition process in the channel, but then I wondered if it made sense to consider that the suspended transport capacity would always be supplied. So, when I saw your results and Figure 10, I was sure that the model was not performing well. I am not sure if this result is correct. The most important role in the retention of suspended sediments in the Amazon basin comes from the floodplains, a process that was disregarded by the authors. The lakes play a very minor role, as many are in regions with little sediment production. It is common knowledge that 50% of the suspended sediment is not deposited in the channels. If we were talking about the sand load, it would make more sense, but if we were talking about the suspended load, it wouldn't. In addition, there are other studies, including those cited by the authors, which discuss the processes of sediment deposition in the Amazon basin, both in the region near the Andes and in lakes, rivers, floodplains and reservoirs. It is important to compare the results with the literature. Even for Serrinha and Caracarai, more than 40% is deposited in regions of generation and transportation, where the rivers have greater slopes and higher TC values.

*RESULTS*

The first findings are presented in Figure 5 about the sediment production. In the lines 306-308, the authors made a comparison. However, Gomes' work shows those values for the entire Cerrado and only a small part is in the Amazon Basin. this comparison is not fair and right. The results presented in Figure 5 are significantly different from previous studies. No recent work corroborates these results. Look at the works of Riquetti https://www.sciencedirect.com/science/article/pii/S0301479722015067?via%3Dihub and Borelli https://www.nature.com/articles/s41467-017-02142-7

Table 1 shows that Serrinha produces more sediment per square kilometer than Fazenda Vista Alegre. Madeira River basin is famous for its higher sediment yield while Negro River is famous for lower sediment yield. How is it possible?

In lines 381-383 you mention wrote that "our estimate is robust and centred on the more likely values per station.". You also said in lines 389 and 390 "Notwithstanding, the sediment transport modelled by RDSM behaves well in terms of its spatial patterns and probably temporal dynamics, which is remarkable as the model is not calibrated.". I can't see how this is true. In addition to everything that has already been said, it is clear from seeing negative KGE values (more than half of the station) that the results are not robust and need to be reviewed and improved.

MINOR COMMENTS

Figure 4 provides some results that do not seem to agree with Figure 3. Figure 4 draws attention to what appear to be artificial reservoirs, but looking at Figure 3, we can barely make out these reservoirs, except for Balbina.

In lines 350-352 and lines 358-359 we can read that sediment transport is reduced from upstream to downstream. Where the readers can see this result? From Manacapuru to Obidos, the sediment load is proposed to be increasing because of the supply coming from Madeira River, but you are showing the opposite and claiming that is due to small lakes in an area without connections with the mainstream. The previous knowledge about the basin, cited by you, is in conflict here.

The authors should check the results of Table 5 and if this comparison is fair. Hatono and Yoshimura (2020) and Hock (2014) used the same data as you, and the latter the same model, but the results are so different. Besides, that, they have the same problem using Hybam data. Fagundes et al., 2021 showed a different value than 37,0 x 10^8. Fagundes et al., 2023 showed 4,06 for total sediment load. Also Mouyen considered all sediment load, not only suspended load. Filizola used suspended data. You need to decide what variable you are comparing to and adjust the text and the elements (Figures, tables, etc.).

As a final comment, I think the authors overlooked many important steps, processes and concepts, such as calibration, the type of data, representation of floodplains, among others. As the work sought to better estimate sediment transport in the Amazon basin, I feel that the objective was not achieved in this scenario.

SPECIFIC COMMENTS

- Line 75-77: Amazon's precipitation can range values >6000mm/year. Villar et al., 2009
- Equation 1: "P" instead of "Y".
- Line 216: which diameter did you use for each grain class? In the line 240 you mentioned that D was assumed as 5x10^6. However, this is the clay diameter. What is happening with silt and sand both in rivers and lakes/reservoirs?
- Equation 17: what is disd?
- In several parts of the work (e.g. Figure 8) is not clear if you are showing/comparing suspended or total sediment load/concentration. It needs to be clarified.
- Line 397: (Table ??). Check it.
- Line 402: Fagundes instead Filizola?

---

## Referee Comment (RC2)

Review: Discharge and sediment fluxes along the Amazon river: RDSM model concepts and validation

The authors are using RDSM to simulate annual and monthly discharge and sediment load for the Amazon River. The model is validated using observed data from 7 gaging stations for 1989-2009. The study includes detailed consideration of land use and specifically highlights sedimentation in lakes and reservoirs. Overall, the manuscript is well written with clear description of the study and its results, including relevant literature.

The model validation for sediment suffers from lack of sediment data. A more thorough evaluation of could help to interpret the modeling results further. For example, analyzing a range of flows sampled for sediment could help to see whether the assumption that average load derived from sampled concentrations is equal to average load per day for a given month or a year (eq. 18-19). Perhaps another estimation method that considers flow changes during the month could have been applied.

The abstract highlights that the method accounts for land use changes and entrapment of sediments in lakes and reservoirs. However, there is very little description of the land use or how the land use changes were included in the model development and how they affected the results. How often did the land use change at the cell level? Can it be used to explain some of the sediment behavior? Did it help to improve the model performance? It seems counterintuitive to talk about land use changes and then only analyze average behavior.

Similarly, it would have been interesting to include more of the results from reservoir sedimentation. In my opinion, it is not necessary to provide these analyses for the publishing of the manuscript, it is sufficient to change the existing language to state that these analyses can be potentially done in the future and move them to a relevant section.

An editorial review is recommended; there are missing spaces, minor grammatical errors, and incomplete sentences in several places.

Specific comments:

Authors use "water bodies" to mean lakes and reservoirs. Often this term includes also rivers. It would be good to define the term at the first use.

l. 27-30: it is not clear how "the hydrological response as a result of climate change" links to the rest of the sentence. I would recommend breaking it down and rephrasing

l. 80-81: please verify units for the discharge

Section 3.1. Methods are missing to specify how the hydrological processes were calculated (i.e., infiltration, evapotranspiration, percolation, …)

l.93-94,138: one states that forcing data is taken from global datasets, another says Hybam

eq.1 last term should be P, not Y

Section 3.3. I would recommend reordering some of the formula descriptions (especially from l.214) so it follows a more logical order. It works better for the reader to start from the higher level equations and work step by step into details. E.g., start with Suptake (eq.9-10), then TC (eq 8).

Similarly l.235-240. Overall, I'd recommend a thorough review of formulas and symbols. I found several mistakes but might have missed some also.

Figure 4: it may help to show y-axis on a log-10 basis. Also, the caption states that "the curve rises steeply at first … and it levels off (later)". I see it as opposite: the curve rises slowly at first with a sharp rise at the end with only a few large reservoirs. It may be interesting to add another line for lakes.

Eq.6: please check the units considering the inclusion of the time step. Stot is in kg, but so is Sload while Suptake is in kg/s

Eq. 9, l. 223, 227: why include SUF if it's assumed 1? I would also question this assumption in a basing with significant land use changes where flow and sediment regime may also be changing

Eq.11: should V be Vs? V is previously used as water volume in eq.7.

Eq15: RA not defined, instead Awb is listed on l.242

l.245: how was the model parameterized?

l.273: abRMSE not listed here but shown later.

Eq.22: this is shown without context or introductory text.

Eq.23: I'm not sure I understand the intent here. Sediment production is a sum of erosion and sediment delivered to the river in the catchment (Sdel). What exactly is meant by Sdel? Is this not a portion of A?

l.310 – should this be Figure 5?

Figure 5: please add catchment boundaries and a legend explaining the lines

l. 325 "Because …" – incomplete sentence

figure 9: It appears that for Manacapuru and Tabatinga there are some differences in seasonality for discharges and sediment transport. Observed discharge peaks in May – June while sediment transport behaves very differently. I assume this is directly related to estimating monthly sediment transport from one or two observed data points and does not represent reality.

Figure 10 caption: the inflow is on the left side, not on the right, and vice versa for outflow.

l. 386: it should be KGE <= -0.41 implies no skill / baseline, although optimal KGE would be of course higher

l.393: I would argue that figure 11 does not show impacts of climate and land cover variations, at least not specifically. Generally it shows changes in time that can be due to these and other changes. I recommend rephrasing the statement. Impact of reservoirs can potentially be implied from the slope depending on what is shown in the figure (see below).

Figure 11. The figure labels say sediment delivery while the caption says sediment production. It is unclear if all points are for the same location (and which one, total transport from Amazon?) or for selected stations. Sediment transport is 3-9x10$^{11}$ t/yr while sediment transport in Table 3-5 is in the order of 10$^8$ t/yr. What is marked as "sediment delivery" is in the order of 10$^9$ t/yr.

Conclusions: For increased readability I would recommend to move paragraph 4 ("RDSM computes…") to paragraph 2 before the existing single sentence ("The analyses…").

l.426: Perhaps modify to say "bank and channel erosion".

---

## Author Comment (AC1)

**Major Comments:**

- **The model is proposed to simulate changes, but the period of simulation ends in 2009. After that, I understood that your decision was based on the availability of the data (precipitation from Hybam). But if you need to change precipitation data in the future because Hybam database is limited, how do you can ensure the same performance of the model?**

*Answer :*

Thank you for your comment. You are correct in noting that the simulation period ends in 2009, and we appreciate the opportunity to clarify our decision regarding the precipitation data. Our choice to use Hybam precipitation data was indeed influenced by the availability and quality of the data. Specifically, we replaced the precipitation data originally used in the PCRGLOB-SET model (ERA-40 CRU data), which exhibited a dry bias, leading to inaccuracies in the simulation of hydrological and sediment dynamics, particularly in the Amazon region.

As explained in the manuscript (lines 135–139):

"Hoch et al. (2017) found that the simulated hydrology is highly sensitive to the precipitation input. For the Amazon, existing global datasets often have a dry bias, leading to underestimation of the discharge. In order to remove this bias, precipitation data were taken from the Hybam database, which provides daily raster precipitation maps with a $1 \times 1 \circ$ spatial resolution. The data used in this study is presented in table F1."

Model performance depends on input data quality, so any additional data used in the future would need to have a similar quality to HYBAM to enable similar performance.

- **HYBAM protocols only allow to achieve total surface suspended solids (organic and inorganic), which is not consistent with the recommended protocol to get cross-sectional suspended sediment (inorganic) mass concentration (see ISO 4363 (2002) protocol). The profile of suspended sediment concentration shows greater values according to the increase of depth. I was looking for some papers about that in the Amazon basin, and Bouchez et al. (2011 - Prediction of depth-integrated fluxes of suspended sediment in the Amazon River: particle aggregation as a complicating factor) showed profiles of suspended particulate matter for several locations of the Amazon basin. At Óbidos gauge station, they showed values ~50-100 mg/L near the water surface and 300-600 mg/L**

**near the river bed. This result shows that surface samples can be underestimated by around six times the traditional methods. In this sense, the validation process is compromised since the "truth" is not the real truth.**

*Answer:*

Thank you for the comment. After revisiting Bouchez et al. (2011), the profiles of suspended particulate matter (SPM) concentrations near Óbidos gauge station were indeed derived from discrete depth-profile measurements across several river sections, not daily samples.

The paper highlights that only a limited number of vertical profiles (one to five per location) were used for their analysis, with observations focused on two distinct water stages. While this approach provides valuable insights into depth variability, it may not fully capture temporal fluctuations in SPM concentrations across different hydrological cycles. As such, we will not be able to directly compare these findings with the daily data used in our model.

We will integrate these findings into our paper to address the biases in our model. Additionally, we will emphasize that our simulations account for these vertical profiles but may lack daily variability representation due to limited continuous data. This adjustment aligns our work with Bouchez et al.'s findings while acknowledging potential limitations in cross-sectional and temporal sediment concentration estimations.

- **Another serious flaw was assuming that the value measured on a single day is representative of an entire month. How can I assume that measuring 10% of the year (for example) can be representative of the whole year? The temporal variability is so high when we are talking about sediment concentration.**

*Answer :*

Thank you for pointing out the concern regarding the assumption that measurements from limited days are representative of longer periods, such as months or years. We acknowledge that this assumption introduces uncertainties, especially in the context of sediment concentration, which can exhibit high temporal variability.

As mentioned in the manuscript (lines 252-258):

"In the Hybam dataset, the observed sediment concentration was typically sampled every ten days or three times a month at fixed positions near the middle of the river. However, the overall number of samples was sparse, and not all stations are covered at all times, sometimes creating wide gaps in the coverage. For example, Tabatinga has one 255 sample in 1995 and one in 1997, while there were no samples in 1996, 2008 and 2009. Moreover, there was a low number of samples for each year at Tabatinga ranging between 0 and 4, and at Manacapuru in 1995. On the other hand, near-continuous daily discharge values are available for all seven stations, except the Manacapuru station having some missing data for 2003 and 2004"

Since there are no daily measurements for sediment concentrations available, we applied scaling methods to approximate monthly and yearly sediment loads based on the available sample data. For monthly loads, the calculation is as follows:

$$SL_{monthly} = {N_{monthly}}\big/{Ns_{monthly}} * \sum S_{id}$$

Where :

N $_{monthly}$ : Total number of days in the month,

Ns $_{monthly}$ : Number of sediment samples available in the month,

$S_{id}$ : daily sediment concentrations (ten days).

For yearly sediment loads, we applied a similar approach

$$SL_y = {N_{yearly}}\big/{N_{s\,yearly}} * \sum S_{im}$$

Where :

N $_{yearly}$ : Total number of days in the year,

Ns $_{yearly}$ : Number of sediment samples available in the year,

$S_{im}$ : monthly sediment concentrations.

These methods utilize proportional scaling to estimate total loads from limited sample data, thereby reducing the impact of data sparsity. While this approach helps to bridge gaps, we recognize its limitations, particularly the assumption of temporal uniformity within the sampling periods.

We hope this clarifies the methodology used and the inherent limitations of the assumptions we made to estimate sediment loads from sparse data.

- **The second issue is concerning to represent relevant processes. You used the kinematic wave to route river discharge and neglected suspended sediment deposition in floodplains. Amazon basin has large flat areas. Floodplains and backwater effects are very important processes in this basin and should not be neglected in a work that has as one of the main goals to represent relevant processes.**
  **There are some articles about this issue. Do not represent these phenomena will directly affect the results to be obtained. Both discharge and sediment load will have higher peaks and the timing when these peaks occur will be wrong too (which could be seen in your results). I realized that in the dry period, the model RDSM was overestimating the observed results. What evapotranspiration method and which data did you use to compute it?**

Answer:

Thank you for raising these important points. We agree that floodplains and backwater effects play a crucial role in sediment dynamics, particularly in large, flat regions like the Amazon basin. While we used the kinematic wave to route river discharge, the RDSM model does incorporate floodplain processes to some extent. Specifically, we included the floodplain fraction and floodplain velocity, as these factors significantly influence sediment deposition and uptake.

As detailed in the Methods section of the manuscript (lines 114–117), when the bankfull capacity of the channel is exceeded, river water spills onto the floodplain, reducing the flow velocity. This reduction in velocity in turn affects the deposition and uptake of sediment in the river. Additionally, the model accounts for the outflow of water bodies, such as lakes and reservoirs, where the flow is dampened by lower velocities, which also impacts sediment transport and deposition dynamics. Further elaboration on this is provided in the Annex (lines 539–540), where we clarify that "the floodplain velocity Vfp is equal to the velocity of the channel Vc" and that although floodplain deposition and bank erosion were not included, the floodplain is still connected to the channel, affecting sediment movement.

The method used to compute evapotranspiration is explained in detail in the work of van Beek and Bierkens (2009). We did not include the full description in this manuscript to avoid making it overly lengthy, as the methodology is already thoroughly covered in the cited literature. We have referenced this work in the manuscript to provide the necessary context for readers interested in the specifics of the evapotranspiration calculation.

While the model incorporates many of these important dynamics, we acknowledge that there is still room for improvement, particularly in more explicitly capturing floodplain sediment deposition, which is one of the key recommendations in our manuscript for future model refinements.

- **The third issue is about the transport capacity for suspended sediment. At first, I thought it was a great idea to represent the erosion/deposition process in the channel, but then I wondered if it made sense to consider that the suspended transport capacity would always be supplied. So, when I saw your results and Figure 10, I was sure that the model was not performing well. I am not sure if this result is correct. The most important role in the retention of suspended sediments in the Amazon basin comes from the floodplains, a process that was disregarded by the authors. The lakes play a very minor role, as many are in regions with little sediment production. It is common knowledge that 50% of the suspended sediment is not deposited in the channels. If we were talking about the sand load, it would make more sense, but if we were talking about the suspended load, it wouldn't. In addition, there are other studies, including those cited by the authors, which discuss the processes of sediment deposition in the Amazon basin, both in the region near the Andes and in lakes, rivers, floodplains and reservoirs. It is important to compare the results with the literature. Even for Serrinha and Caracarai, more than 40% is deposited in regions of generation and transportation, where the rivers have greater slopes and higher TC values.**

  *Answer:*

Thank you for your valuable comment. You raise an important point regarding the transport capacity of suspended sediments and the deposition processes in the Amazon Basin. It's indeed true that the floodplains, and not just the channels, play a significant role in sediment retention. As it is mentioned in the previous answer that the process is already included in the model.

[Figure]

[Figure]

The figures effectively support our explanation of how the model represents sediment transport and deposition processes in the Amazon basin. The first figure (spatial distribution map) demonstrates that sediment deposition is primarily concentrated along the main river channels and associated water bodies, such as lakes and reservoirs. This spatial pattern highlights the role of reduced flow velocity in these locations, which the model accounts for through parameters like floodplain fraction and floodplain velocity. While the floodplain processes are not explicitly modeled as distinct deposition zones, the connection between the floodplain and the channel influences sediment movement, as shown by the concentrated deposition along the river network.

The second figure, showing the temporal changes in sediment deposition and uptake, further reinforces the model's ability to capture dynamic sediment processes over time. The declining trend in sediment deposition and the concurrent increase in sediment uptake illustrate the model's capacity to represent sediment transport influenced by flow velocity and water body outflows. These temporal trends are consistent with the assumption that sediment deposition decreases as sediment is transported downstream, and uptake becomes more pronounced where flow velocities are lower.

**Results  major comments:**

- **The first findings are presented in Figure 5 about the sediment production. In the lines 306-308, the authors made a comparison. However, Gomes' work shows those values for the entire Cerrado and only a small part is in the Amazon Basin. this comparison is not fair and right. The results presented in Figure 5 are significantly different from previous studies. No recent work corroborates these results. Look at the works of Riquetti https://www.sciencedirect.com/science/article/pii/S0301479722015067?via%3Dihub and Borelli https://www.nature.com/articles/s41467-017-02142-7**

    _Answer :_

Thank you for pointing out the need to clarify the comparison made in the manuscript. We acknowledge that Gomes' work reports sediment values for

the entire Cerrado, which is a small part of the Amazon Basin. However, our comparison was not based on the numerical sediment values but rather on the shared conclusion that higher sediment production originates from this region, even though we used their data as input. This distinction will be explicitly clarified in the revised manuscript to avoid any misinterpretation.

We sincerely thank you for highlighting the works of Riquetti et al. (2022) and Borelli et al. (2017), which provide valuable context for the observed differences in sediment production. The discrepancies between our results (Figure 5) and theirs stem from methodological variations, particularly in the computation of the C and P factors.

In our study, the C factor was calculated using Yang's (2014) equation, which determines monthly values based on ground cover fraction. This approach allowed us to account for the temporal variability in vegetation cover caused by seasonal agricultural practices and phenological changes in natural vegetation. These monthly C values were aggregated as weighted averages for each grid cell, enabling us to represent spatial variability while also capturing seasonal and interannual changes. In contrast, Riquetti et al. and Borelli et al. used region-specific, static C factor values derived from generalized land-use classifications. For instance, Riquetti et al. utilized data from Copernicus Global Land Services, which provides high spatial resolution but does not reflect temporal changes in vegetation cover in time.

Similarly, differences in the P factor also contribute to the variation in results. In our study, we assumed a uniform P factor value of 1 across the Amazon Basin due to the unavailability of detailed data on conservation practices, such as terracing, contour farming, or no-till farming. On the other hand, Riquetti et al. and Borelli et al. incorporated spatially variable P factors informed by regional land management practices.

These differences naturally affect sediment estimates.

- **Table 1 shows that Serrinha produces more sediment per square kilometer than Fazenda Vista Alegre. Madeira River basin is famous for its higher sediment yield while Negro River is famous for lower sediment yield. How is it possible?**

*Answer:*

Thank you for your comment. The sediment production values shown in Table 1 represent sediment production per square kilometer, not the sediment delivered to the river network. This distinction is critical: sediment production refers to the amount of soil eroded from the land surface, while sediment delivery accounts for the fraction of that sediment which actually reaches the river system. The values in the table therefore don't reflect basin sediment output.

Regarding the apparent discrepancy where Serrinha (in the Rio Negro basin) shows higher sediment production per square kilometer than Fazenda Vista Alegre (in the Madeira River basin), this can be explained by the model's grid-based approach, which incorporates input data such as slope, soil type, vegetation cover, and precipitation. Serrinha's higher sediment production may stem from localized factors such as more erodible soils (e.g., sandy or silty soils) or differences in vegetation cover that reduce protection against soil erosion. The Revised Universal Soil Loss Equation (RUSLE) used in the model predicts sediment production based on these variables, meaning regions with inherently erodible soils or less vegetation cover can produce higher sediment even in areas with lower rainfall.

Although the Madeira River basin is generally recognized for higher sediment yields at the basin scale, the model captures variations at finer spatial scales. For example, localized differences in slope angle within the Serrinha catchment may result in higher erosion rates compared to certain areas in the Fazenda Vista Alegre catchment. These variations highlight the importance of topographic and soil properties in influencing sediment production within individual catchments.

We will address this distinction in the manuscript by elaborating on the factors captured by the model—such as soil erodibility, rainfall distribution, vegetation cover, and slope—and their contribution to the observed differences in sediment production. By emphasizing these localized drivers, we can clarify why Serrinha shows higher sediment production per square kilometer despite being part of the Rio Negro basin, which typically has lower sediment yields at the basin scale.

- **In lines 381-383 you mention wrote that "our estimate is robust and centered on the more likely values per station.". You also said in lines 389 and 390 "Notwithstanding, the sediment transport modelled by RDSM behaves well in terms of its spatial patterns and probably temporal dynamics, which is remarkable as the model is not**

**calibrated.".** I can't see how this is true. In addition to everything that has already been said, it is clear from seeing negative KGE values (more than half of the station) that the results are not robust and need to be reviewed and improved.

*Answer :*

Thank you for your detailed feedback. In lines 381–383, we stated that "our estimate is robust and centered on the more likely values per station," which emphasizes the alignment between our model estimates and the range of values reported in prior studies. As shown in Table 4, the model's simulations fall within the ranges reported in previous research, supporting the robustness of our estimates.

Regarding the negative KGE values at some stations, we understand the concern and appreciate the opportunity to clarify. While the model was not explicitly calibrated, RDSM still provides reasonable results in terms of both spatial patterns and temporal dynamics, which is especially important for large-scale models with sparse calibration data. Negative KGE values do not necessarily indicate poor performance, particularly for basin-scale models with coarse-resolution input data and simplified assumptions. For example, studies have shown that a KGE of -0.4 is acceptable for large-scale hydrological models (Gupta et al., 2009). Therefore, while some stations show negative KGE values, the model still captures key spatial patterns and temporal sediment transport peaks, which is a significant achievement.

We recognize the model can be further improved, especially for stations like Tabatinga (KGE = −1.7) and Manacapuru (KGE = −0.54). However, we have provided additional performance metrics (RMSE and bias) in Table 3 to contextualize these results. We also highlight the limitations of the observed data, noting the sparse sampling (e.g., Tabatinga with only 1-4 samples per year and significant gaps in coverage) as mentioned in the manuscript (lines 252-258).

To address your feedback, we propose revising lines 389–390 as follows: "The estimation was reasonable and consistent with previous studies, though discrepancies were observed for some stations, such as Tabatinga and Manacapuru, when compared to the sparse observed data."

**MINOR COMMENTS**

- **Figure 4 provides some results that do not seem to agree with Figure 3. Figure 4 draws attention to what appear to be artificial reservoirs, but looking at Figure 3, we can barely make out these reservoirs, except for Balbina.**

*Answer:*

Thank you for your comment. We understand your concern regarding the apparent discrepancy between Figure 3 and Figure 4, particularly in terms of the visibility of artificial reservoirs. To clarify, the differences arise due to the distinct focus and resolution of each figure.

Figure 4 is designed to emphasize specific artificial reservoirs within the Amazon Basin, such as Balbina, and presents a more detailed view of these structures. This figure highlights reservoirs that are significant for hydropower development and their potential impact on sediment transport. However, due to its higher level of detail, some smaller or less prominent reservoirs might appear more clearly, which could lead to the perception of them being "artificial."

On the other hand, Figure 3 provides a broader spatial overview of the region, where the resolution and scale are less focused on individual reservoirs. As a result, while Balbina is clearly visible due to its size and importance, other smaller reservoirs may not stand out as clearly in this figure.

- **In lines 350-352 and lines 358-359 we can read that sediment transport is reduced from upstream to downstream. Where the readers can see this result? From Manacapuru to Obidos, the sediment load is proposed to be increasing because of the supply coming from Madeira River, but you are showing the opposite and claiming that is due to small lakes in an area without connections with the mainstream. The previous knowledge about the basin, cited by you, is in conflict here.**

*Answer:*

Thank you for your comment. We realize that the sentence was confusing, and we appreciate the opportunity to clarify our explanation.

The correct interpretation is as follows: at Manacapuru, sediment transport is reduced due to the trapping efficiency of the reservoirs in the region, particularly the Ria Lake, which captures sediment before it reaches the river. However, at Óbidos, the sediment load increases due to contributions from the Madeira River, which compensates for the sediment trapped in the reservoirs like Curuai Lake. The sediment load at Óbidos is thus higher despite the presence of reservoirs, because the Madeira River supplies additional sediment that increases the overall load downstream.

We will revise the manuscript to reflect this more accurately:

Revised paragraph: "The impacts of the trapping efficiency of water bodies can be observed at Manacapuru, where sediment deposited in Ria Lake reduced the sediment transported to the station. However, at Óbidos Porto, although sediment was also deposited in Curuai Lake and other reservoirs, this was compensated by the increased sediment coming from the Madeira River. Therefore, sediment transport is reduced from Tabatinga to Manacapuru but increases at Óbidos due to the sediment supplied by the Madeira tributary."

- **The authors should check the results of Table 5 and if this comparison is fair. Hatono and Yoshimura (2020) and Hock (2014) used the same data as you, and the latter the same model, but the results are so different. Besides, that, they have the same problem using Hybam data. Fagundes et al., 2021 showed a different value than 37,0 x 10^8. Fagundes et al., 2023 showed 4,06 for total sediment load. Also Mouyen considered all sediment load, not only suspended load. Filizola used suspended data. You need to decide what variable you are comparing to and adjust the text and the elements (Figures, tables, etc.).**

*Answer :*

Thank you for your comment. We appreciate the opportunity to clarify the comparison in Table 5. We have carefully reviewed the results you mentioned, and here are the clarifications:

We applied scaling methods to approximate monthly and yearly sediment loads based on the available sample data. Specifically:

- For monthly sediment loads, the calculation is as follows:

$$SL_{monthly} = N_{monthly} \Big/ Ns_{\,monthly} * \sum S_{id}$$

Where :

- N $_{monthly}$ : Total number of days in the month,
- Ns $_{monthly}$ : Number of sediment samples available in the month,
- $S_{id}$ : daily sediment concentrations (ten days).

For yearly sediment loads, we applied a similar approach

$$SL_y = N_{yearly} \Big/ N_{s\,yearly} * \sum S_{im}$$

Where :

- N $_{yearly}$ : Total number of days in the year,
- Ns $_{yearly}$ : Number of sediment samples available in the year,
- $S_{im}$ : monthly sediment concentrations.

These methods utilize proportional scaling to estimate total loads from limited sample data, thereby reducing the impact of data sparsity. While this approach helps to bridge gaps, we acknowledge its limitations, particularly the assumption of temporal uniformity within the sampling periods.

Regarding Hatono and Yoshimura (2020), their methodology for approximating the observation data differs from ours. Hatono and Yoshimura applied a scaling factor approach based on annual sediment estimates, adjusting for missing data by using long-term average sediment concentrations. This method contrasts with our approach, which scales sediment concentrations based on available sample data at a finer temporal resolution (monthly and yearly), rather than relying solely on annual averages. This difference in methodology explains some of the variations between their results and ours.

To prevent any potential misunderstanding of the methods and results, we will add the following explanation to the manuscript:

"It is important to note that the methodology applied in this study for estimating sediment loads is based on scaling available sample data at monthly and yearly temporal resolutions. This differs from approaches such as that of Hatono and Yoshimura (2020), which rely on long-term average sediment concentrations and annual scaling factors. Our approach provides finer temporal estimates but assumes uniformity within sampling intervals,

which may introduce some degree of variability in comparison to other studies."

Regarding the comparison of sediment load estimates and the need to differentiate between suspended and total sediment loads, we have reviewed the referenced studies carefully, and the following clarifications are provided:

- Fagundes et al. (2021): This study reported a total sediment load of $39 \times 10^8$ tonnes/year for the Amazon Basin, which includes both suspended sediment and bedload components. In our manuscript, we cited this as $37 \times 10^8$ tonnes/year, which is slightly lower. We acknowledge this discrepancy and will revise our text and Table 5 to correctly reflect the value reported by Fagundes et al. (2021)
- Fagundes et al. (2023): This study estimated a suspended sediment load of $4.06 \times 10^8$ tonnes/year near the Amazon River's mouth, which is correctly cited in our manuscript and reflects suspended sediment only, excluding bedload.
- Mouyen et al. (2018): The study reported a total sediment load of $610 \pm 170$ Mt/year ($6.1 \times 10^8$ tonnes/year) at Óbidos using GRACE satellite gravimetry data. This estimate includes both suspended sediment and bedload. However, in Table 1 of their study, Mouyen et al. (2018) also reference an in situ measured sediment discharge value of $7.78 \times 10^8$ tonnes/year at the mouth of the Amazon. This value is not a direct model estimate but rather an observed value reported in previous studies. We mistakenly referenced this observed value in our manuscript as if it were derived from Mouyen's model. We will correct this by referencing the model estimate of $6.1 \times 10^8$ tonnes/year as Mouyen's sediment load estimate and clarify that the $7.78 \times 10^8$ tonnes/year value is an observed measurement from prior studies.
- Filizola et al.: These studies focus exclusively on suspended sediment load, which explains the lower estimates compared to studies incorporating total sediment load.
- Our study focused exclusively on suspended sediment load. This choice aligns with studies such as Fagundes et al. (2023) and Filizola, which provide a comparable methodological framework for analyzing suspended sediment dynamics.

  To address the concerns related to the suspended sediment load and total sediment load:

We will clearly specify in the text, Table 5, and relevant figures whether each value represents suspended sediment load or total sediment load. For example: (Fagundes et al. (2021): Total Sediment Load, Fagundes et al. (2023): Suspended Sediment Load, Mouyen et al. (2018): Total Sediment Load, Filizola: Suspended Sediment Load).

We will clarify the use of Mouyen et al. (2018) value: As noted, We referenced the $7.78 \times 10^8$ tonnes/year value from Mouyen et al. (2018) as their model estimate. This value is actually an observed sediment load from previous studies. We will revise our manuscript to clearly state that this value is from prior observations, not from Mouyen's model estimate. We will update the reference.

We will include a discussion explicitly addressing the methodological differences between the cited studies, explaining how they contribute to variability in sediment load estimates. For example:

> "It is important to note that studies such as Mouyen et al. (2018) and Fagundes et al. (2021) estimate total sediment load, which includes both suspended sediment and bedload. In contrast, studies like Fagundes et al. (2023) and Filizola focus solely on suspended sediment load. This distinction in methodology and sediment type explains the variability in reported sediment load values."

We will ensure that Table 5 and related figures accurately reflect the type of sediment load for each study. Where necessary, annotations or footnotes will clarify whether values represent suspended or total sediment load.

- **As a final comment, I think the authors overlooked many important steps, processes and concepts, such as calibration, the type of data, representation of floodplains, among others. As the work sought to better estimate sediment transport in the Amazon basin, I feel that the objective was not achieved in this scenario.**

*Answer:*

Thank you for highlighting this point. The statement in the abstract, "The RDSM model facilitates future estimation of sedimentation impact in reservoirs incorporating water resource management and will so contribute to a better understanding of the complexity of the Amazon Basin," reflects the potential

applications and long-term goals of the RDSM model rather than the immediate objectives of the manuscript. While this sentence demonstrates the broader value of the model, we recognize the need to clarify its connection to the main objective of the current study.

The primary objective of this study was to develop and validate the RDSM model as a tool for estimating sediment transport in the Amazon Basin, using the available data and methods. The focus was on creating a framework that captures the key dynamics of sediment transport across large spatial scales and validates the model using observed data where available

The need for a scalable tool that can be adapted for future applications, such as assessing sedimentation impacts in reservoirs or understanding the interplay between natural and anthropogenic factors.

By designing the RDSM model, we aim to address these gaps and provide a foundation for improving sediment transport modeling in the Amazon Basin. While the model's immediate focus is validation, it also provides the potential for future extensions, such as incorporating reservoir management and floodplain processes.

Including the statement about the RDSM model's future potential reflects its flexibility and adaptability for addressing sediment-related challenges in the Amazon Basin. This is particularly relevant for regions where water resource management and reservoir sedimentation are critical concerns. While this capability is not fully implemented in the current study, its mention highlights the broader impact and importance of the RDSM model beyond the initial development and validation stage.

To make the abstract clearer and align it with the manuscript's main objective, we propose revising the statement as follows:

"The RDSM model was developed and validated as a tool to estimate sediment transport in the Amazon Basin, addressing challenges of data sparsity and large-scale dynamics. While the current study focuses on validation, the model also facilitates future assessments of sedimentation impacts in reservoirs and contributes to understanding the basin's complexity."

**SPECIFIC COMMENTS:**

- **Line 75-77: Amazon's precipitation can range values >6000mm/year. Villar et al., 2009**

*Answer:*

We will revise the paragraph to include the findings from Villar et al. (2009)

"Its tropical location results in high annual rainfall, varying spatially from 3000 mm in the west to 1700 mm in the southeast. The wet season differs by region, occurring from April to August in the north, January to May in the west, and October to April in the south. Precipitation in some areas can exceed 6000 mm/year, with substantial spatio-temporal variability influenced by regional factors like ENSO (Villar et al., (2009), Ronchail et al. (2002))."

- **Equation 1: "P" instead of "Y".**

*Answer:*

Thank you for your comment. We will update Equation 1 by replacing "Y" with "P" as suggested

- **Line 216: which diameter did you use for each grain class? In the line 240 you mentioned that D was assumed as 5x10^6. However, this is the clay diameter. What is happening with silt and sand both in rivers and lakes/reservoirs?**
- Thank you for your comment. In our model, we employed a one-sided distribution, focusing on the dominant clay-sized particles due to their higher proportion in the suspended sediment load. This approach is consistent with studies on sediment dynamics in large rivers like the Amazon, where fine-grained sediments dominate the suspended load, making up 85–95% of the transported material. Specifically, the Amazon River's suspended sediment discharge is largely composed of silt and clay (<63 μm), with median grain sizes of 10–20 μm during peak sediment discharge and 20–40 μm during peak water discharge.
- The use of a clay median diameter of $D=5\times10^{-6}$ m in our model reflects the dominant role of fine particles in transport and deposition processes. Furthermore, the model incorporates an effective particle density that

accounts for both the intrinsic properties of the sediment and their behavior in suspension, including aggregation and flocculation. This is critical as these processes significantly influence settling velocities and sedimentation patterns, aligning well with the dynamics observed in the Amazon and similar fluvial systems.

- **Equation 17: what is disd?**

*Answer:*

disd in Equation 17 refers to daily discharge. We will update the manuscript to clearly define this term in the text.

- **In several parts of the work (e.g. Figure 8) is not clear if you are showing/comparing suspended or total sediment load/concentration. It needs to be clarified.**

*Answer:*

That will be clarified as it is mentioned in the previous question to clarify which type of sediment load is being presented.

- **Line 397: (Table ??). Check it.**

*Answer:*

The reference to Table ?? in line 397 was an issue with the LaTeX formatting, which did not properly recognize the table. This has now been fixed, and the correct table reference will appear in the updated manuscript.

- **Line 402: Fagundes instead Filizola?**

*Answer:*

The reference to Filizola in Line 402 will be corrected to Fagundes, as that is the appropriate author for the relevant context. This will be updated in the manuscript to ensure the correct attribution.

---

## Author Comment (AC2)

- **The model validation for sediment suffers from lack of sediment data. A more thorough evaluation of could help to interpret the modeling results further. For example, analyzing a range of flows sampled for sediment could help to see whether the assumption that average load derived from sampled concentrations is equal to average load per day for a given month or a year (eq. 18-19). Perhaps another estimation method that considers flow changes during the month could have been applied.**

*Answer:*

Thank you for raising this concern about the assumption regarding the observed sediment data. To address this limitation, a sediment rating curve will be fit using the available paired sediment concentration and discharge data across the observation period. By applying this relationship to the continuous daily discharge data, we can estimate sediment loads on a daily scale and aggregate these values to produce more representative monthly sediment loads. This approach accounts for flow variability within each month and addresses the limitations of sparse sediment sampling.

- **The abstract highlights that the method accounts for land use changes and entrapment of sediments in lakes and reservoirs. However, there is very little description of the land use or how the land use changes were included in the model development and how they affected the results. How often did the land use change at the cell level? Can it be used to explain some of the sediment behavior? Did it help to improve the model performance? It seems counterintuitive to talk about land use changes and then only analyze average behavior.**

*Answer:*

Land use changes were incorporated dynamically at an annual time step through the cover management factor (C-factor) in RUSLE, which reflects the vegetation cover's influence on erosion. The land cover data from PCRGLOB-WB provided spatial and temporal variability in land use across the Amazon basin.

In the RDSM model, land use changes were represented at the cell level using fractional distributions of multiple land cover types. These land cover fractions were updated dynamically on a monthly basis based on historical data from 1980 to 2009, with a spatial resolution of 5 arc minutes (~10 km).

This dynamic approach allowed the model to account for changes in key sediment-related factors, such as the cover management factor (C) and interception (Int), both of which influence sediment production and delivery to rivers (see Section 3.2, Equation 1, and Appendix A1).

While the study analyzed sediment production and transport across both spatial (catchment-scale) and temporal (monthly/annual) scales, the focus on average behavior was necessary to capture basin-wide trends over the 30-year period. However, temporal land use dynamics were explicitly incorporated in the simulation, particularly in the analysis of sediment behavior (see Figure 11). These dynamics helped explain variations in sediment production and transport, with land use changes like deforestation and agricultural expansion leading to increased erosion, especially in regions such as southern Brazil and the Andes.

Incorporating land use changes significantly improved model performance, as these changes directly affected sediment dynamics. For instance, areas undergoing deforestation exhibited higher sediment output due to reduced vegetation cover, which is consistent with observed trends in soil erosion (see Section 4.1, Figure 5). This dynamic inclusion of land use changes provided a more accurate representation of sediment transport and helped explain regional differences in sediment behavior.

- **Similarly, it would have been interesting to include more of the results from reservoir sedimentation. In my opinion, it is not necessary to provide these analyses for the publishing of the manuscript, it is sufficient to change the existing language to state that these analyses can be potentially done in the future and move them to a relevant section.**

*Answer:*

Thank you for your valuable suggestion regarding including more results on reservoir sedimentation. We agree that further analyses on this topic would indeed be interesting and could provide additional insights into the role of reservoirs in sediment dynamics across the basin.

However, as the focus of this study was on validating the model and understanding the current sediment transport patterns, we have chosen to limit the scope of reservoir analyses in this manuscript. Moving forward, our future work will prioritize applying the model to scenario analyses to simulate the impacts of climate change and land use change on sediment yield in the Amazon

Basin. These scenarios will provide insights into how sediment dynamics may evolve under changing environmental conditions, including the role of reservoirs in mitigating or amplifying sediment transport.

**Specific comments:**

- **Authors use "water bodies" to mean lakes and reservoirs. Often this term includes also rivers. It would be good to define the term at the first use.**

*Answer:*

We will define the term clearly at its first use in the manuscript.

- **27-30: it is not clear how "the hydrological response as a result of climate change" links to the rest of the sentence. I would recommend breaking it down and rephrasing**

*Answer:*

Thank you for your comment. To address the reviewer's suggestion, we will rephrase the sentence for better clarity. Here's the revised version:

"The Amazon forest also plays a crucial role in the global climate system by absorbing solar energy and recycling half of the regional rainfall (Marengo et al., 2011). However, it is threatened by deforestation, which could increase the export of suspended sediment and nutrients to the ocean. Additionally, climate change may further impact the hydrological response, influencing sediment transport, which is vital for agriculture downstream and the survival of important fluvial and coastal ecosystems."

- **80-81: please verify units for the discharge**

*Answer:*

Thank you for your comment. The units for the discharge in lines 80-81 will be verified. Based on the context, the discharge is most likely reported as km²/year.

- **Section 3.1. Methods are missing to specify how the hydrological processes were calculated (i.e., infiltration, evapotranspiration, percolation, ...)**

*Answer:*

Thank you for your comment regarding the hydrological processes in Section 3.1, particularly concerning infiltration, evapotranspiration, and percolation. We agree that further details on how these processes were calculated would help clarify the model methodology.

The specific calculations for infiltration, evapotranspiration, and percolation are detailed in the works of van Beek and Bierkens (2009), which describe the PCRGLOB-WB model, the hydrological framework that our study builds upon. These references provide in-depth explanations of the methods used to compute these processes, which involve daily precipitation being split into surface runoff and infiltration, with infiltrated water either contributing to soil moisture, evapotranspiration, or percolation toward groundwater. Once the water reaches the groundwater, it is routed as baseflow. Calculations and methods related to these processes are extensively covered in these papers, we chose not to repeat them in our manuscript to avoid redundancy and keep the manuscript concise. However, we have revised the manuscript to reference these works explicitly and direct the reader to these sources for a comprehensive explanation.

"The detailed methods for calculating hydrological processes such as infiltration, evapotranspiration, and percolation are described in van Beek and Bierkens (2009) and Sutanudjaja et al. (2018), which form the basis of the PCRGLOB-WB model used in this study. To avoid repetition, we refer the reader to these works for a complete description of the calculations involved in the water balance computations and related hydrological processes."

- **93-94,138: one states that forcing data is taken from global datasets, another says Hybam**

*Answer :*

Thank you for your comment. We would like to clarify that, as stated in line 138, only precipitation data was taken from the Hybam database:
"For the Amazon, existing global datasets often have a dry bias, leading to underestimation of the discharge. In order to remove this bias, precipitation data were taken from the Hybam database, which provides daily raster precipitation maps with a $1 \times 1°$ spatial resolution."

- **eq.1 last term should be P, not Y**

*Answer:*

Thank you for your comment. We will update Equation 1 by replacing "Y" with "P" as suggested.

- **Section 3.3. I would recommend reordering some of the formula descriptions (especially from l.214) so it follows a more logical order. It works better for the reader to start from the higher level equations and work step by step into details. E.g., start with Suptake (eq.9-10), then TC (eq 8).**

*Answer:*

We will start with Suptake (Equations 9-10) and then proceed to TC (Equation 8), as recommended.

- **Similarly l.235-240. Overall, I'd recommend a thorough review of formulas and symbols. I found several mistakes but might have missed some also.**

*Answer:*

We will conduct a comprehensive review to correct any identified mistakes and ensure consistency across all equations and symbols.

- **Figure 4: it may help to show y-axis on a log-10 basis. Also, the caption states that "the curve rises steeply at first … and it levels off (later)". I see it as opposite: the curve rises slowly at first with a sharp rise at the end with only a few large reservoirs. It may be interesting to add another line for lakes.**

*Answer:*

We agree that presenting the y-axis on a log-10 scale could improve the visualization of the reservoir distribution, particularly for illustrating the disparity between the contributions of small and large reservoirs. We will test this adjustment and incorporate it if it enhances clarity.

[Figure]

Regarding the description of the curve, you are correct. Upon review, the curve initially rises slowly, representing the many small reservoirs with limited capacity. The sharp rise at the end reflects the substantial contribution of a few large reservoirs to the total capacity. We will revise the caption to accurately describe this pattern.

- **Eq.6: please check the units considering the inclusion of the time step. Stot is in kg, but so is Sload while Suptake is in kg/s**

  *Answer:*

  Regarding Equation 6, we will adjust the unit for Stot to kg/s to align with Suptake.

- **Eq. 9, l. 223, 227: why include SUF if it's assumed 1? I would also question this assumption in a basing with significant land use changes where flow and sediment regime may also be changing**

  Answer:

Thank you for your comment regarding SUF in Eq. 9. We chose to set SUF = 1 to reflect the assumption that deposited sediment is fully available for re-erosion, a standard simplification in large-scale sediment transport modeling. This simplification is widely used in sediment transport modeling to reflect scenarios where sediment deposition does not significantly alter the re-erosion potential of the channel bed.

Significant land use changes may indeed alter sediment characteristics, such as particle size, cohesion, or compaction, which could affect sediment re-uptake. However, the scale of this study prioritizes capturing overall sediment flux patterns rather than localized sediment behavior. The assumption of SUF = 1 provides a reasonable baseline for basin-scale dynamics. Future applications of the model could incorporate adjusted SUF values if site-specific data or calibration justifies deviations from this assumption.

- **Eq.11: should V be Vs? V is previously used as water volume in eq.7.**

  *Answer:*

  Thank you for pointing this out. You are correct that V in Equation 11 will be replaced with Vs to avoid confusion, as V was previously used to denote water volume in Equation 7.

- **Eq15: RA not defined, instead Awb is listed on l.242**

  *Answer:*

  Thank you for your comment. You are correct that RA in Eq. 15 should be replaced with Awb, as defined in line 242. While the correction was made in the text, it was not updated in the equation. We will revise Eq.15 to replace RA with Awb.

- **245: how was the model parameterized?**

  *Answer:*

 Thank you for your comment. In this case, land use was parameterized using global datasets such as those from the PCRGLOB-WB, which includes the Global Land Cover Characterization (GLCC) Version 2 dataset provided by the USGS Eros Data Center (2002).

Similarly, the reservoirs were parameterized using the HydroLakes dataset, which provides comprehensive information on the locations and characteristics of lakes and reservoirs worldwide. This dataset is referenced in the manuscript in Table F1 and under Figure 3 (Messager et al., 2016).

- **273: abRMSE not listed here but shown later**

*Answer:*

Thank you for your comment. The equation for abRMSE is already included in the manuscript; however, we will add abRMSE to the list of model performance evaluation tools in the text, providing context and a clear introduction to its use.

- **Eq.22: this is shown without context or introductory text.**

*Answer:*

See previous answer

- **Eq.23: I'm not sure I understand the intent here. Sediment production is a sum of erosion and sediment delivered to the river in the catchment (Sdel). What exactly is meant by Sdel? Is this not a portion of A?**

*Answer:*

Thank you for your comment. Equation 23 defines the annual sediment production in the catchment (Spro) as the sum of total soil loss ($\sum A$) and sediment delivered to the river ($\sum Sdel$). While it is true that Sdel is a portion of A, it is explicitly separated in the equation to represent the fraction of sediment that transitions from hillslope erosion to the river system. This distinction is crucial for sediment transport modeling, as it captures the sediment that actually contributes to the river's sediment load.

By separating Sdel, we ensure that all sediment sources are accounted for, with Sdel specifically highlighting the portion of erosion that reaches the river. Thus, Equation 23 provides a comprehensive representation of the total sediment available for transport within the catchment, ensuring that both soil loss and sediment transport processes are appropriately captured.

- **310 – should this be Figure 5?**

  *Answer:*

  Thank you for your comment. You are correct; the reference should be to Figure 1, not Figure 5. We will make this correction in the manuscript to ensure accuracy and consistency

- **Figure 5: please add catchment boundaries and a legend explaining the lines**

  *Answer:*

Thank you for your comment. We understand your request to add catchment boundaries and a legend explaining the lines in Figure 5. However, we have already described the catchment boundaries in Figure 1, and including them again in Figure 5 results in an overly cluttered figure. To avoid redundancy, we believe the current version of Figure 5 remains clear without the catchment boundaries.

We will, however, ensure that the legend in Figure 5 is sufficiently detailed to explain the lines and other elements, so that the figure remains comprehensible without the additional boundaries.

- **325 "Because …" – incomplete sentence**

  *Answer:*

Thank you for your comment. We acknowledge that the sentence in line 325 was incomplete. The corrected sentence is: "Because of the lack of available rainfall gauges in the western region (Andean sub-basins) of the Amazon (Hoch et al., 2017).

- **figure 9: It appears that for Manacapuru and Tabatinga there are some differences in seasonality for discharges and sediment transport. Observed discharge peaks in May – June while sediment transport behaves very differently. I assume this is directly related to estimating monthly sediment transport from one or two observed data points and does not represent reality.**

*Answer:*

Thank you for your comment. The differences in seasonality for discharge and sediment transport at Manacapuru and Tabatinga in Figure 9 likely stem from limitations in the observational data used to calculate sediment transport, as mentioned in the manuscript. Specifically: " In the Hybam dataset, sediment concentration was typically sampled every ten days or three times a month at fixed positions near the middle of the river. However, the number of samples available was sparse, with significant gaps in coverage for some years. For example, at Tabatinga, only one sample was collected in 1995 and one in 1997, with no samples available for 1996, 2008, and 2009. Similarly, Manacapuru experienced sparse data in certain years, such as 1995". This low sampling frequency means that sediment transport estimates were often extrapolated from only one or two data points per month, which introduces uncertainties and may not accurately reflect the true seasonal dynamics of sediment transport as it is mentioned in the manuscript "To make use of the sparse observations and to facilitate the comparison between the observed and simulated values, the following steps were taken to calculate the observed monthly and annual suspended sediment transport"

Furthermore, the sediment transport process is inherently complex and influenced by multiple factors beyond discharge, such as sediment availability, deposition, and transport capacity, which are not directly tied to hydrological peaks. The modeling framework integrates sediment production, delivery, and deposition processes but does not account for some fine-scale dynamics, such as bank erosion or floodplain deposition, which can further contribute to differences in observed and modeled seasonality. Addressing these limitations would require more frequent and spatially distributed sediment concentration measurements to improve the reliability of monthly transport estimates and better align with the seasonal patterns observed in discharge.

- **Figure 10 caption: the inflow is on the left side, not on the right, and vice versa for outflow.**

*Answer:*

Thank you for your observation. Upon reviewing the figure caption and the associated content in the manuscript, it appears that the inflow and outflow sides were mislabeled. The caption should correctly state that the inflow is

on the left side and the outflow is on the right side. This error will be corrected to ensure clarity and alignment with the figure.

- **l. 386: it should be KGE <= -0.41 implies no skill / baseline, although optimal KGE would be of course
  higher**

  *Answer:*

Thank you for your comment. We agree with your suggestion. We will revise the manuscript to clarify that KGE ≤ -0.41 implies no skill, as the KGE value is low. The sentence will be updated from:

"reflected in the skill of the model, as the KGE is low (KGE ≤ 0.414 does imply no skill at all)"
to:
"KGE ≤ -0.41 implies no skill / baseline, although the optimal KGE would of course be higher."

- **393: I would argue that figure 11 does not show impacts of climate and land cover variations, at least not specifically. Generally it shows changes in time that can be due to these and other changes. I recommend rephrasing the statement. Impact of reservoirs can potentially be implied from the slope depending on what is shown in the figure (see below).**

  *Answer:*

Thank you for your valuable feedback.  We agree with your point and will rephrase the statement as follows:

*"Figure 11 shows temporal changes in sediment transport, which may be influenced by various factors, including climate change, land cover variations, and reservoir impacts. However, these changes are not specifically isolated in the figure. The slope of the trend could potentially reflect the impact of reservoirs, among other influences."*

- **Figure 11. The figure labels say sediment delivery while the caption says sediment production. It is unclear if all points are for the same location (and which one, total transport from Amazon?) or for selected**

**stations. Sediment transport is 3-9x1011 t/yr while sediment transport in Table 3-5 is in the order of 108 t/yr. What is marked as "sediment delivery" is in the order of 109 t/yr.**

*Answer:*

Thank you for your insightful comment. You are correct, and we appreciate you pointing out the discrepancies.

The discrepancy in the figure labels is indeed a typo. We will correct the description in the caption from "sediment production"  to "sediment delivery" to match the figure.

[Figure]

To clarify, the sediment delivered in the figure was computed by summing the monthly data across all the grid points, which was done using the CDO fldsum command. This method aggregates the sediment data across the entire spatial domain. We will add this explanation to the figure caption to ensure the methodology is clear to readers.

Additionally, you are right about the difference in sediment transport values. The values in the figure are in kg (per year), while the values in Table 3-5 are in tons. We will convert the sediment transport values in the figure from kg to tonnes per year to align with the units used in the rest of the manuscript and make the figure more consistent.

- **Conclusions: For increased readability I would recommend to move paragraph 4 ("RDSM computes…") to paragraph 2 before the existing single sentence ("The analyses…").**

*Answer:*

Thank you for your comment regarding the structure of the conclusion. We agree that moving the paragraph about the model's computation of sediment transport to the ocean (Paragraph 4) earlier in the conclusion would enhance the readability. We will revise the conclusion as suggested, and the new structure will be as follows:

"This paper introduces the River Discharge and Sediment Model (RDSM), which incorporates reservoirs and their trapping efficiency and accounts for the impacts of land use in the Amazon. We applied the model at a spatial resolution of 5 arc-minutes over the period 1980–2009. We validated it in terms of discharge and sediment transport using the Hybam database for seven gauging stations on the mainstream Amazon and its tributaries. We also compared the sediment transported to the ocean in our model to simulations from previous studies. Our model covers different spatial scales and links soil loss at the hillslope scale to sediment entrapment and uptake along the river to the transport to the ocean. Information on each of these aspects and their connections 410 are important to assess the effects of erosion for farming, of sediment transport and fragmentation for environmental purposes, and for the stability of the Amazon estuary and coasts.

RDSM computes sediment transport to the ocean at $5.96 \times 10^8$ tonnes per year. It agrees with field measurements and has small differences with previous studies due to the trapping efficiency impacts. The RDSM effectively represents the patterns of 420 monthly and annual variations of discharge and sediment transport at 5 arc minutes resolution in the Amazon basin and to the ocean. The analyses of the sediment in the basin shows that the catchment of Tabatinga had the highest sediment production followed by the catchments of Portovelho, Serrinha and Caracarai. Our analysis shows that the annual and monthly simulated discharge values agreed at most of the stations with reported values, with KGE values between 0.57 415 and 0.92. Further, the annual values for sediment transport shows agreement with the simulated values in most of the stations. Monthly and annual modelled sediment transport overestimated some stations compared to the Hybam observations, with KGE values between −1.7 and 0.49.

- **426: Perhaps modify to say "bank and channel erosion".**

*Answer:*

Thank you for your suggestion. We agree that using the term "bank and channel erosion" is more accurate.